# The wiring diagram of a glomerular olfactory system

Matthew E Berck[1,2†], Avinash Khandelwal[3,4†], Lindsey Claus[1,2], Luis Hernandez-Nunez[1,2], Guangwei Si[1,2], Christopher J Tabone[1‡], Feng Li[5], James W Truman[5], Rick D Fetter[5], Matthieu Louis[3,4*], Aravinthan DT Samuel[1,2*], Albert Cardona[5*]

[1]Department of Physics, Harvard University, Cambridge, United States; [2]Center for Brain Science, Harvard University, Cambridge, United States; [3]EMBL-CRG Systems Biology Program, Centre for Genomic Regulation, The Barcelona Institute of Science and Technology, Barcelona, Spain; [4]Universitat Pompeu Fabra, Barcelona, Spain; [5]Janelia Research Campus, Howard Hughes Medical Institute, Ashburn, United States

**Abstract** The sense of smell enables animals to react to long-distance cues according to learned and innate valences. Here, we have mapped with electron microscopy the complete wiring diagram of the *Drosophila* larval antennal lobe, an olfactory neuropil similar to the vertebrate olfactory bulb. We found a canonical circuit with uniglomerular projection neurons (uPNs) relaying gain-controlled ORN activity to the mushroom body and the lateral horn. A second, parallel circuit with multiglomerular projection neurons (mPNs) and hierarchically connected local neurons (LNs) selectively integrates multiple ORN signals already at the first synapse. LN-LN synaptic connections putatively implement a bistable gain control mechanism that either computes odor saliency through panglomerular inhibition, or allows some glomeruli to respond to faint aversive odors in the presence of strong appetitive odors. This complete wiring diagram will support experimental and theoretical studies towards bridging the gap between circuits and behavior.

*For correspondence: mlouis@ crg.eu (ML); samuel@physics. harvard.edu (ADTS); cardonaa@ janelia.hhmi.org (AC)

†These authors contributed equally to this work

Present address: ‡Fly Base, Harvard University, Cambridge, United States

Competing interests: The authors declare that no competing interests exist.

## Introduction

An animal uses its sense of smell to navigate odor gradients, and to detect the threat or reward associated with an odor. In the nervous system, odorants are detected by olfactory receptor neurons (ORNs) whose axons organize centrally into glomeruli by olfactory receptor type (*Wang et al., 1998*; *Vosshall et al., 2000*). Uni- and multi-glomerular projection neurons (PNs) relay olfactory inputs to higher-order brain areas (*Stocker et al., 1990*; *Liang et al., 2013*). Common between mammals and insects (*Vosshall and Stocker, 2007*; *Su et al., 2009*), PNs target two major brain centers, one associated with learning and memory (such as the mushroom bodies (MB) in insects), and another that mediates some innate behaviors (such as the lateral horn (LH) in insects) (*Fischbach and Heisenberg, 1984*; *Heisenberg et al., 1985*; *Stocker et al., 1990*; *Sosulski et al., 2011*). Local neurons (LNs) mediate communication between glomeruli, implementing computations such as gain control (*Olsen and Wilson, 2008*). While the connectivity of a few glomeruli has been recently partially reconstructed in the adult fly (*Rybak et al., 2016*), the complete number and morphology of cell types and the circuit structure with synaptic resolution is not known for any glomerular olfactory system.

In the *Drosophila* larva, we find a similarly organized glomerular olfactory system of minimal numerical complexity (*Figure 1a*). In this tractable system, each glomerulus is defined by a single, uniquely identifiable ORN (*Fishilevich et al., 2005*; *Masuda-Nakagawa et al., 2009*), and almost all neurons throughout the nervous system are expected to be uniquely identifiable and stereotyped

**eLife digest** Our sense of smell can tell us about bread being baked faraway in the kitchen, or whether a leftover piece finally went bad. Similarly to the eyes, the nose enables us to make up a mental image of what lies at a distance. In mammals, the surface of the nose hosts a huge number of olfactory sensory cells, each of which is tuned to respond to a small set of scent molecules. The olfactory sensory cells communicate with a region of the brain called the olfactory bulb. Olfactory sensory cells of the same type converge onto the same small pocket of the olfactory bulb, forming a structure called a glomerulus. Similarly to how the retina generates an image, the combined activity of multiple glomeruli defines an odor.

A particular smell is the combination of many volatile compounds, the odorants. Therefore the interactions between different olfactory glomeruli are important for defining the nature of the perceived odor. Although the types of neurons involved in these interactions were known in insects, fish and mice, a precise wiring diagram of a complete set of glomeruli had not been described. In particular, the points of contact through which neurons communicate with each other – known as synapses – among all the neurons participating in an olfactory system were not known.

Berck, Khandelwal et al. have now taken advantage of the small size of the olfactory system of the larvae of *Drosophila* fruit flies to fully describe, using high-resolution imaging, all its neurons and their synapses. The results define the complete wiring diagram of the neural circuit that processes the signals sent by olfactory sensory neurons in the larva's olfactory circuits.

In addition to the neurons that read out the activity of a single glomerulus and send it to higher areas of the brain for further processing, there are also numerous neurons that read out activity from multiple glomeruli. These neurons represent a system, encoded in the genome, for quickly extracting valuable olfactory information and then relaying it to other areas of the brain.

An essential aspect of sensation is the ability to stop noticing a stimulus if it doesn't change. This allows an animal to, for example, find food by moving in a direction that increases the intensity of an odor. Inhibition mediates some aspects of this capability. The discovery of structure in the inhibitory connections among glomeruli, together with prior findings on the inner workings of the olfactory system, enabled Berck, Khandelwal et al. to hypothesize how the olfactory circuits enable odor gradients to be navigated. Further investigation revealed more about how the circuits could detect slight changes in odor concentration regardless of whether the overall odor intensity is strong or faint. And, crucially, it revealed how the worst odors – which can signal danger – can still be perceived in the presence of very strong pleasant odors.

With the wiring diagram, theories about the sense of smell can now be tested using the genetic tools available for *Drosophila*, leading to an understanding of how neural circuits work.

(*Manning et al., 2012*; *Vogelstein et al., 2014*; *Li et al., 2014*; *Ohyama et al., 2015*). Some of the olfactory LNs and PNs have already been identified (*Masuda-Nakagawa et al., 2009*; *Thum et al., 2011*; *Das et al., 2013*). This minimal glomerular olfactory system exhibits the general capabilities of the more numerically complex systems. For example, as in other organisms (*Friedrich and Korsching, 1997*; *Nagayama et al., 2004*) and in the adult fly (*Bhandawat et al., 2007*; *Nagel and Wilson, 2011*; *Kim et al., 2015*), the output of the uniglomerular PNs tracks the ORN response (*Asahina et al., 2009*), which represents both the first derivative of the odorant concentration and the time course of the odorant concentration itself (*Schulze et al., 2015*). Like in the adult fly (*Olsen and Wilson, 2008*) and zebrafish (*Zhu et al., 2013*), gain control permits the larval olfactory system to operate over a wide range of odorant concentrations (*Asahina et al., 2009*). The olfactory behaviors exhibited by the larva have been well studied (mostly in 2nd and 3rd instar larvae), in particular chemotaxis (*Cobb, 1999*; *Bellmann et al., 2010*; *Gomez-Marin et al., 2011*; *Gershow et al., 2012*; *Schulze et al., 2015*; *Gepner et al., 2015*; *Hernandez-Nunez et al., 2015*), as well as the odor tuning and physiological responses of ORNs (*Fishilevich et al., 2005*; *Louis et al., 2008*; *Asahina et al., 2009*; *Kreher et al., 2008*; *Montague et al., 2011*; *Mathew et al., 2013*). Additionally the larva presents odor associative learning (*Gerber and Stocker, 2007*). Obtaining the wiring diagram of all neurons synaptically connected to the ORNs would enable the formulation of system-

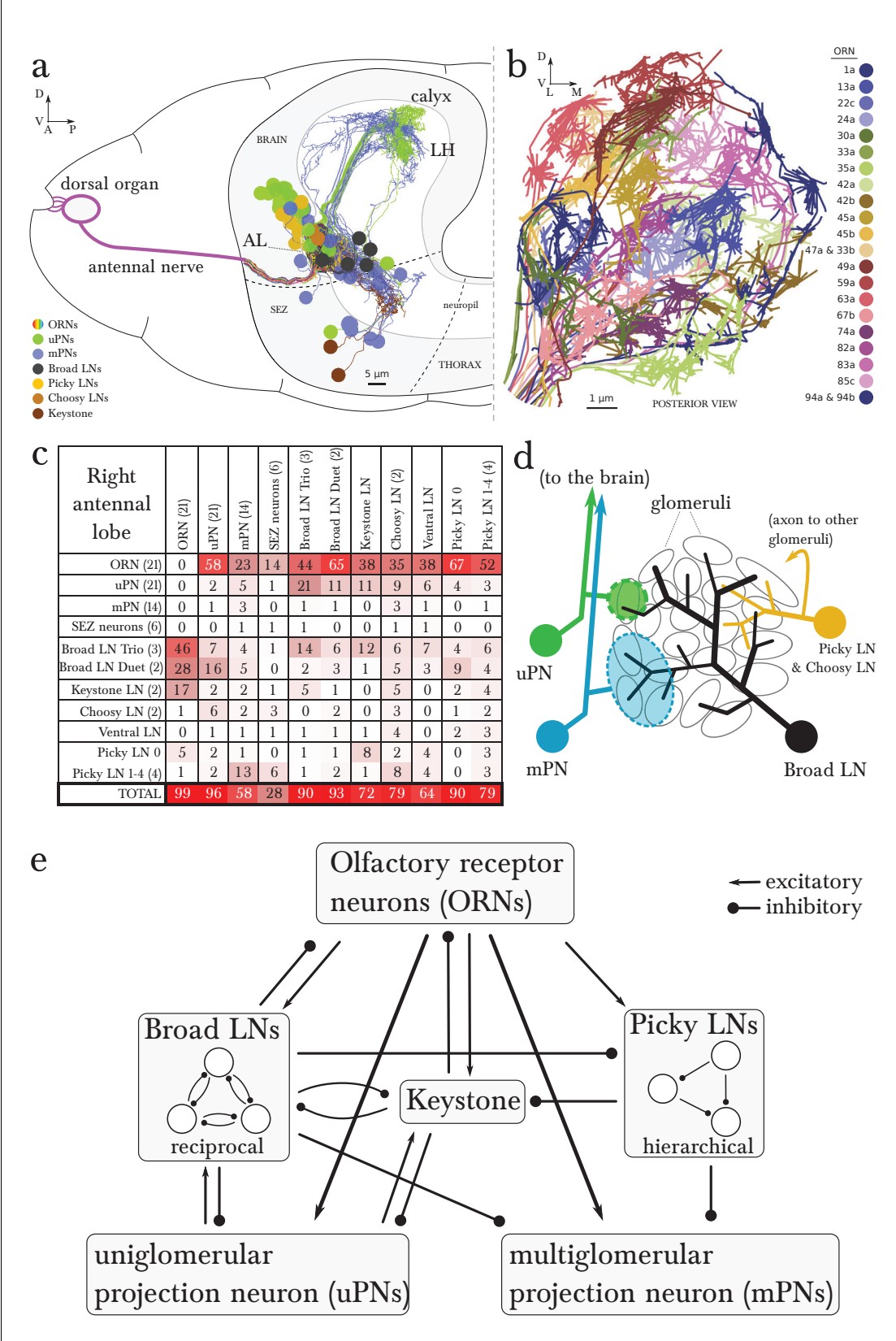

**Figure 1.** Overview of the wiring diagram of the glomerular olfactory system of the larval *Drosophila*. (a) Schematic of the olfactory system of the larval *Drosophila* with EM-reconstructed skeletons overlaid. The ORN cell bodies are housed in the dorsal organ ganglion, extend dendrites into the dome of

*Figure 1 continued on next page*

*Figure 1 continued*

the dorsal organ, and emit axons to the brain via the antennal nerve. Like in all insects, neuron cell bodies (circles) reside in the outer layer of the nervous system (grey), and project their arbors into the neuropil (white) where they form synapses. Also shown are the major classes of local neurons (Broad LNs, Picky LNs and Keystone) and the 2 classes of projection neurons (uPNs and mPNs). The arbors of the Broad LNs (black) specifically innervate the AL. LNs and mPN dendrites can extend into the subesophageal zone (SEZ), innervated by sensory neurons of other modalities. uPNs project to specific brain areas (mushroom body calyx and lateral horn; LH), and mPNs mostly project to other nearby brain areas. (**b**) The larva presents 21 unique olfactory glomeruli, each defined by a single ORN expressing a single or a unique pair of olfactory receptors. We reconstructed each ORN with a skeleton and annotated its synapses, here colored like the skeleton to better illustrate each glomerulus. See *Figure 1—figure supplement 1* for individual renderings that aided in the identification of each unique ORN. (**c**) Summary connectivity table for the right antennal lobe with all major neuron classes (4 neuromodulatory neurons and the descending neuron from the brain were omitted), indicating the percent of postsynaptic sites of a column neuron contributed by a row neuron. For most neurons, the vast majority of their inputs originates in other neurons within the antennal lobe. In parentheses, the number of neurons that belong to each cell type. We show only connections with at least two synapses, consistently found among homologous identified neurons in both the left and right antennal lobes. Percentages between 0 and 0.5 are rounded down to 0. (**d**) Schematic of the innervation patterns of the main classes of LNs and PNs in the antennal lobe. White ovals represent the glomeruli. Solid circles are cell bodies. Shaded areas with dotted outlines represent the extent of the PN dendritic arbors, with each uPN (green) innervating one glomerulus and each mPN (blue) innervating multiple glomeruli. Their axons (arrows) project to the brain. Broad LNs (black) are axonless and present panglomerular arbors. Picky LN (orange) dendrites span multiple glomeruli and their axons (arrow; not shown) target a different yet overlapping set of glomeruli as well as regions outside the olfactory system. Choosy LNs are similar to the Picky LNs but their axons remain within the antennal lobe. (**e**) A simplified wiring diagram of the larval olfactory system with only the main connections. ORNs are excitatory. All shown LNs are inhibitory. Broad LNs reciprocally connect to all glomeruli and each other and thus engage in presynaptic inhibition (on ORNs) and postsynaptic inhibition (on uPNs). Picky LNs form a hierarchical circuit and selectively synapse onto mPNs. Another LN, Keystone, receives inputs from ORNs, one Picky LN and non-ORN sensory neurons, and can potentially alter the operational mode of the entire olfactory system by altering the pattern of inhibition (see text).

The following figure supplements are available for figure 1:

**Figure supplement 1.** Electron microscopy view of the antennal lobe of *Drosophila* larva.

**Figure supplement 2.** A single, identified ORN for each glomerulus in the antennal lobe of the first instar larva.

**Figure supplement 3.** EM-reconstructed arbors of all LNs.

level hypotheses of olfactory circuit function to explain the observed behavioral and functional properties. The reduced numerical complexity and dimensions of the larval olfactory system, the similarity of its organization and capabilities to other organisms, and the tractability of the larva as a transparent genetic model organism, make it an ideal model system in which to study the complete circuit architecture of a glomerularly organized olfactory processing center.

We reconstructed from electron microscopy all synaptic partners of the 21 ORNs for both the left and right antennal lobes of a first instar larva (*Figure 1b*; *Figure 1—figure supplement 1* and *2*). Per side, we found 21 uniglomerular PNs (uPNs; one per glomerulus), 14 LNs, 14 multiglomerular PNs (mPNs), 4 neuromodulatory neurons, 6 subesophageal zone (SEZ) interneurons and 1 descending neuron (*Figure 1c,d*). These identified neurons present stereotyped connectivity when comparing the left and right antennal lobes. The lack of undifferentiated neurons in the 1st instar antennal lobe, and comparisons with light-microscopy images of other instars suggests that the 1st instar antennal lobe contains all the neurons present throughout larval life. Here, we analyze this complete wiring diagram on the basis of the known function of circuit motifs in the adult fly and other organisms and known physiological properties and behavioral roles of identified larval neurons. We found two distinct circuit architectures structured around the two types of PNs: a uniglomerular system where each glomerulus participates in a repeated, canonical circuit, centered on its uPN (*Python and Stocker, 2002*); and a multiglomerular system where all glomeruli are embedded in structured, hetereogeneous circuits read out by mPNs (*Figure 1e*; [*Das et al., 2013*]). We also found that the inhibitory LNs structure a circuit that putatively implements a bistable inhibitory system. One state could compute odor saliency through panglomerular lateral inhibition, that is, by suppressing the less active glomeruli in favor of the more active ones. The other may enable select glomeruli, specialized for aversive odors, to respond to faint stimuli in a background of high, appetitive odor stimuli. We discuss the role of these two possible operational states and how neuromodulatory neurons and brain feedback neurons participate in the interglomerular circuits.

## Results

### Neurons of the olfactory system of *Drosophila* larva

We mapped the wiring diagram of the first olfactory neuropil of the larva by reconstructing the left and right ORNs and all their synaptic partners. We used a complete volume of the central nervous system (CNS) of a first instar larva, imaged with serial section electron microscopy ([*Ohyama et al., 2015*]; see Materials and methods for online image data availability; *Figure 1—figure supplement 1*). We reconstructed 160 neuronal arbors using the software CATMAID (*Saalfeld et al., 2009*; *Schneider-Mizell et al., 2016*). All together, the 160 neurons add up to a total of 38,684 postsynaptic sites and 55 millimeters of cable, requiring about 600,000 mouse clicks over 736 hr of reconstruction and 431 hr of proofreading. Only 136 of 14,346 (0.9%) postsynaptic sites of ORNs remained as small arbor fragments (comprising a total of 0.25 millimeters of cable, or 0.5% of the total reconstructed) that could not be assigned to any neuron.

We sorted the 160 reconstructed neurons into 78 pairs of bilaterally homologous neurons and 4 ventral unpaired medial (VUM) neurons (2 are mPNs and 2 are octopaminergic 'tdc' neurons; [*Selcho et al., 2014*]). These 78 pairs we further subdivided into 21 pairs of ORNs, 21 pairs of uPNs, 13 pairs of mPNs (plus 2 additional VUM mPNs), 14 pairs of LNs, 6 pairs of neurons projecting to the SEZ ('SEZ neurons'), 1 pair of descending neurons from the brain, 1 pair of serotonergic neurons (CSD; [*Roy et al., 2007*]), and 1 pair of octopaminergic non-VUM neurons ('lAL-1'; [*Selcho et al., 2014*]).

The 14 pairs of LNs originate in 5 different lineages (*Figure 1—figure supplement 3*). We assigned the same name to neurons of the same lineage, and numbered each when there is more than one per lineage. LNs connect to other neuron classes stereotypically in the two antennal lobes (*Figure 2*). We selected names reminiscent of either their circuit role or anatomical feature, including 'Broad' to refer to panglomerular arbors; 'Picky' and 'Choosy' for LNs of two different lineages (and different neurotransmitter; see below) with arbors innervating select subsets of glomeruli; 'Keystone' for a single pair that mediate interactions between LNs of different circuits; and 'Ventral LN' for a single pair of LNs with ventral cell bodies. We also determined the neurotransmitters of LNs that were previously unknown (*Figure 2—figure supplement 1*). We introduce the properties of each LN type below with the olfactory circuits that they participate in.

### The uniglomerular system

In vertebrates and most arthropods, olfactory glomeruli are defined by a group of same-receptor ORNs converging onto a set of glomerular-specific PNs (mitral and tufted cells in the mouse and zebrafish olfactory bulb) (*Stocker et al., 1990*; *Satou, 1992*; *Ressler et al., 1994*; *Wang et al., 1998*; *Distler and Boeckh, 1996*). In *Drosophila* larva, this system is reduced to a single ORN and a single uPN per glomerulus (*Python and Stocker, 2002*; *Ramaekers et al., 2005*; *Masuda-Nakagawa et al., 2009*). With one exception (35a, which has 2 bilateral uPNs), our EM-reconstructed wiring diagram is in complete agreement with these findings (*Figures 1b*, *3a*). Most of the larval uPNs project to both the MB and the LH (*Figure 3a*), like in the adult fly (*Stocker et al., 1990*; *Luo et al., 2010*).

### Circuits for interglomerular inhibition

In insects (adult fly, bee, locust) and in vertebrates, the excitation of glomeruli is under control of inhibitory LNs that mediate functions such as gain control, which define an expanded dynamic range of uPN responses to odors (*Sachse and Galizia, 2002*; *Lei et al., 2002*; *Olsen and Wilson, 2008*; *Olsen et al., 2010*; *Zhu et al., 2013*). We found that most non-sensory inputs to the larval uPNs (*Figure 1c*) are from a set of 5 panglomerular, axonless, and GABAergic (*Thum et al., 2011*) neurons that we named Broad LNs (*Figure 3b,c*; *Figure 3—figure supplement 1*). These 5 Broad LNs also account for most inputs onto the ORN axons (*Figure 1c*), therefore being prime candidates for mediating both intra- and interglomerular presynaptic inhibition (onto ORNs) as observed in the adult fly with morphologically equivalent cells (*Wilson and Laurent, 2005*; *Olsen and Wilson, 2008*; *Olsen et al., 2010*; *Chou et al., 2010*), and in the larva (*Asahina et al., 2009*).

We divided the 5 Broad LNs into two classes, Trio and Duet, based on the number of neurons of each type (*Figure 3b*). While both types provide panglomerular presynaptic inhibition (onto ORN axons), the Duet makes far more synapses onto the dendrites of the uPNs. This may indicate a far

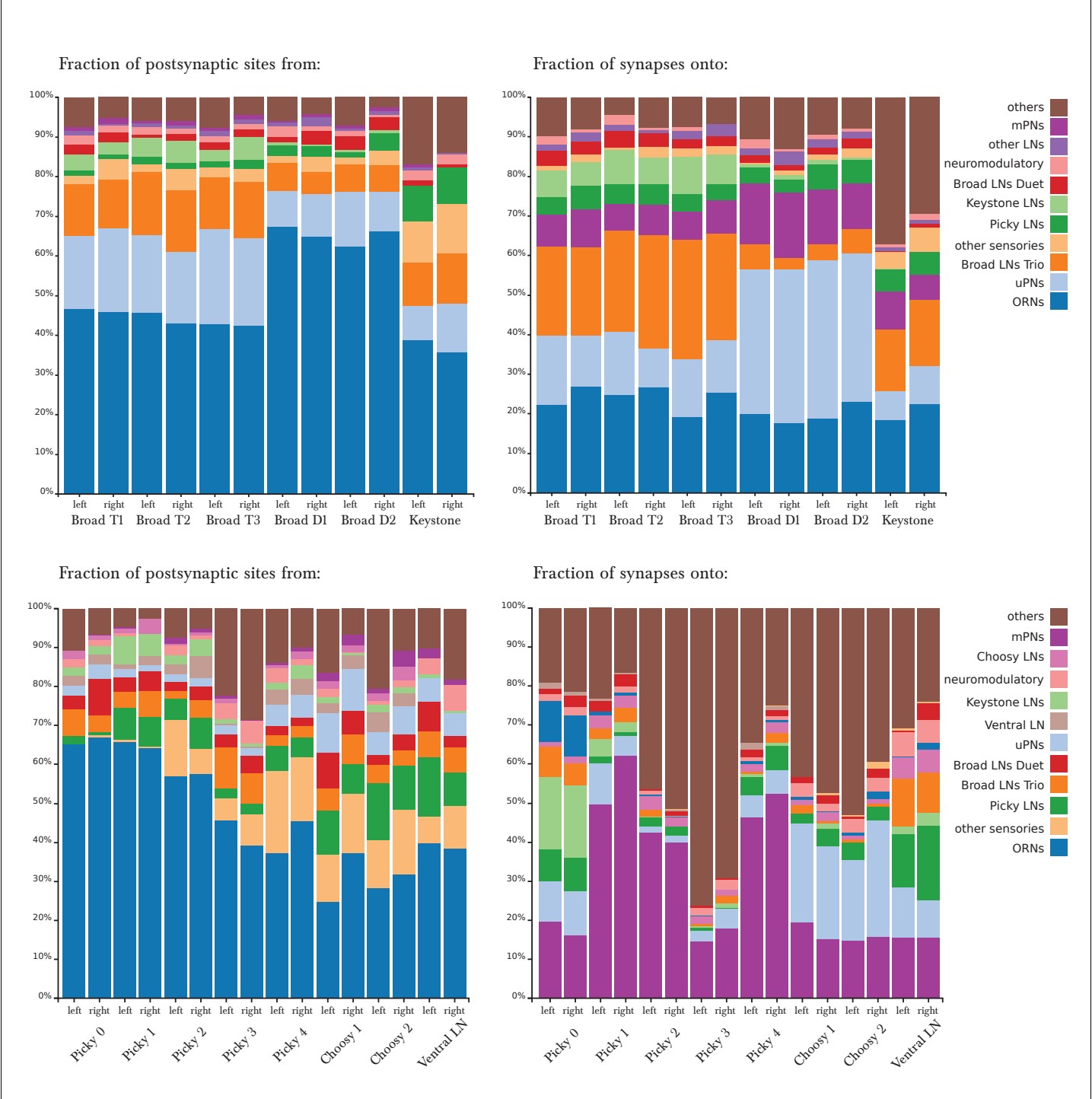

**Figure 2.** Percentage of synapses of LNs from/onto specific cell types. The entry for each neuron presents two bars, for the left and right homologs. Top row, Broad LNs and Keystone. T: Trio, D: Duet. *Left,* differences between the Trio and Duet subtypes are evident in the fraction of inputs that originates in ORNs, uPNs and Keystone. The Duet subtype presents a far larger fraction of its inputs from ORNs, and barely receives any inputs from Keystone. By its pattern of inputs, Keystone resembles a Broad LN Trio neuron, except for the large fraction of non-ORN inputs and the inputs from Picky LNs (specifically from Picky LN 0). *Right,* note how the Trio subtype devote about 25% of their synapses to each other, whereas the Duet subtype preferentially targets uPNs, providing postsynaptic inhibition to the glomeruli (both lateral and feedforward inhibition). Keystone differs from the Broad LNs in that it targets uPNs much more weakly, preferring instead the Broad LN Trio and a variety of other neurons. Bottom row, Picky LNs, Choosy LNs and Ventral LN. *Left,* the fraction of inputs from ORNs stands out as a large difference among Picky LNs, with Picky LN 3 and 4 receiving substantially fewer, similarly to Choosy LNs. The fraction of inputs received from other Picky LNs (green) is among the most distinguishing feature of Picky LN 0,

*Figure 2 continued on next page*

*Figure 2 continued*
which receives close to none. *Right*, in contrast to the similar patterns of inputs onto all Picky LNs, Picky LN 0 stands out as very different from other Picky LNs in its choice of downstream synaptic partners, spreading approximately evenly between ORNs, uPNs, mPNs, other Picky LNs and Keystone. Choosy LNs strongly prefer uPNs, being therefore strong providers of postsynaptic inhibition to glomeruli. Notice that Picky LNs, Choosy LNs and Ventral LN have a larger fraction of synapses to/from 'others', with their arbors spreading towards adjacent sensory neuropils in the SEZ.
The following figure supplement is available for figure 2:

**Figure supplement 1.** Neurotransmitters of Keystone LN and Picky LNs.

stronger role for the Broad LN Duet in postsynaptic inhibition (onto uPN dendrites; *Figure 3d*). In the adult fly, presynaptic inhibition implements gain control (*Olsen and Wilson, 2008*), and postsynaptic inhibition plays a role in uPNs responding to the change in ORN activity (*Nagel et al., 2015*; *Kim et al., 2015*). The two types of glomerular inhibition are provided by two separate cell types, and may therefore be modulated independently. For example, the uPNs emit dendritic outputs that primarily target the Broad LN Trio (*Figure 3d*), indicating that the output of the glomerulus contributes more to presynaptic than to postsynaptic inhibition. Similar excitatory synapses from uPNs to inhibitory LNs have been shown in vertebrates (*Rall et al., 1966*), and synapses from PNs to LNs and vice versa have been described in the adult fly (*Rybak et al., 2016*).

Beyond their role in pre- and postsynaptic inhibition of ORNs and uPNs respectively, the Broad LNs synapse onto all neurons of the system, including other LNs and mPNs (*Figure 1c*, *Figure 2*). Therefore Broad LNs may be defining a specific dynamic range for the entire antennal lobe, enabling the system to remain responsive to changes in odorant intensities within a wide range. Importantly, Broad LNs also synapse onto each other (*Figure 3c,e*) like in the adult (*Okada et al., 2009*; *Rybak et al., 2016*). Furthermore, the two types of Broad LNs have a different ratio of excitation and inhibition, originating in the preference of Trio to synapse far more often onto each other than onto Duet (*Figure 3e*, *Figure 2*). This suggests that the two types not only have different circuit roles but also have different properties.

Another GABAergic cell type, that we call the Choosy LNs (two neurons; *Figures 1c*, *3c*, *Figure 1—figure supplement 3*), contributes exclusively to postsynaptic inhibition for most glomeruli. Unlike the Broad LNs, Choosy LNs have a clear axon innervating most glomeruli, while their dendrites collect inputs from only a small subset of glomeruli (*Figure 3c*; *Figure 1—figure supplement 3*; *Figure 3—figure supplement 1*). Therefore some glomeruli can drive postsynaptic inhibition of most glomeruli. Additionally, the inputs from Choosy LNs tend to be more proximal to the axon initial segment of the uPNs (*Gouwens and Wilson, 2009*) unlike those of Broad LNs which are more uniformly distributed throughout the uPN dendritic arbor (*Figure 3—figure supplement 2*). In the adult, ORNs tend to synapse at the most distal PN dendritic terminals, allowing for some LN inhibition to occur via synapses more proximal to the axon initial segment (*Rybak et al., 2016*). This pattern of spatially structured inputs suggests that different inhibitory LN types may exert different effects on uPN dendritic integration.

## The multiglomerular system

Parallel to the uniglomerular readout by the 21 uPNs, we found 14 multiglomerular PNs (mPNs; *Figure 4a*). Each mPN receives unique and stereotyped inputs from multiple ORNs (*Figure 4c*) or at least from one ORN and multiple unidentified non-ORN sensory neurons in the SEZ (*Figure 4a*). The mPNs originate in multiple neuronal lineages and project to multiple brain regions; most commonly the lateral horn (LH) but also regions surrounding the MB calyx. Of the 14 mPNs, three project to the calyx itself (mPNs b-upper, b-lower and C2) and another (mPN cobra) to the MB vertical lobe (*Figure 4a*). In addition to the 14 mPNs that project to the brain, we identified an extra 6 oligoglomerular neurons that project to the SEZ (SEZ neurons; *Figure 1c*; *Figure 4—figure supplement 1*). A class of mPNs has been described in the adult fly (*Liang et al., 2013*) but their projection pattern does not match any of the larval mPNs. In strong contrast to uPNs, mPNs are very diverse in their lineage of origin, their pattern of inputs, and the brain areas they target. A small subset of mPNs has been identified via light microscopy before (*Thum et al., 2011*; *Das et al., 2013*).

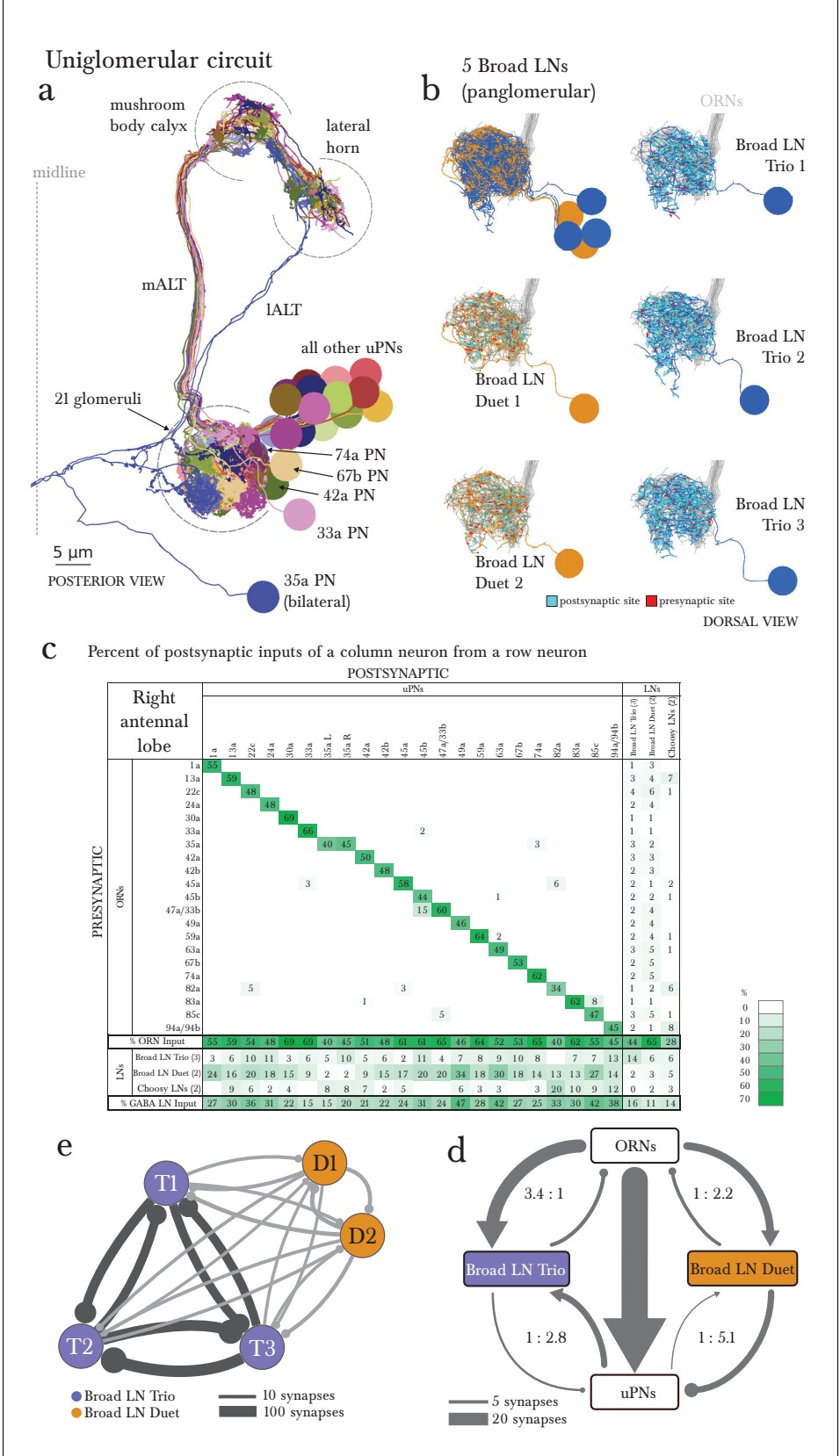

**Figure 3.** The uniglomerular circuit consists of 21 glomerular-specific projection neurons, which interact primarily with their corresponding ORN and with the 5 panglomerular LNs (Broad LNs), each an identified neuron. (a)
*Figure 3 continued on next page*

*Figure 3 continued*

Posterior view of the EM-reconstructed uPNs of the right antennal lobe. The dendrites of each uPN delineate the glomerular boundaries, and the axons project to both the mushroom body (required for learning and memory; [*Michels et al., 2011*]) and the lateral horn (implicated in innate behaviors; [*de Belle and Heisenberg, 1994*]). 19 uPNs are likely generated by the same neuroblast lineage BAmv3 (*Das et al., 2013*) (although the uPNs for 42a, 74a, and 67b are slightly separated from the rest), and the other two (the uPNs for 33a and 35a) clearly derive from two other neuroblasts. Notice that the 35a uPN is bilateral, ascends through a different tract, and receives additional inputs outside of the antennal lobe. The 33a uPN does not synapse within the calyx and the 82a uPN does not continue to the lateral horn. The left antennal lobe (not shown) is a mirror image of the right one. (b) Dorsal view of the EM-reconstructed, axonless Broad LNs (Duet in orange; Trio in blue) shown together and individually. All neurons are on the same lineage: BAlc (*Das et al., 2013*). The pre-(red) and post-(cyan) synaptic sites on these panglomerular neurons are fairly uniformly distributed. ORNs in grey for reference. These neurons extend posteriorly out of the olfactory glomeruli to receive synapses from 2 non-ORN sensory neurons that enter the brain via the antennal nerve. (c) Percentage of the total number of postsynaptic sites on the dendrite of an uPN, Broad LN or Choosy LN (columns) that originate in a given ORN or LN (rows) for the right antennal lobe. Since the 35a uPN is bilateral, we include inputs to it from both antennal lobes. We show only connections with at least two synapses, consistently found among homologous identified neurons in both the left and right antennal lobes. Percentages between 0 and 1 are rounded to 1, but totals are computed from raw numbers. The uniglomerular nature of uPNs (notice the green diagonal) and panglomerular nature of Broad LNs is evident. The Broad LN Duet generally contributes more synapses onto uPNs than the Broad LN Trio does. While the number of synapses that an ORN makes onto its uPN varies widely (24–120 synapses; see *Supplementary file 1* and *2*), this number is tailored to the size of the target uPN dendrite given that percentage of inputs the ORN contributes to the uPN is much less varied (mostly 45–65%). For an extended version of this table that includes all LNs, see *Figure 3—figure supplement 1*. (d) Both Broad LN types (Trio and Duet) mediate presynaptic inhibition (synapses onto ORN axons) similarly, but the Duet shows far stronger postsynaptic inhibition (synapses onto uPN dendrites) while the Trio receives far more dendro-dendritic synapses from uPNs. Connections among Broad LNs are not shown for simplicity. Each arrow is weighted linearly by the number of synapses for an average single Broad LN of each type. (e) The 5 Broad LNs that govern this circuit synapse reciprocally, with the Trio type synapsing more strongly onto each other. Shown here for the right antennal lobe with arrow thickness weighted by the square root of the number of synapses.

The following figure supplements are available for figure 3:

**Figure supplement 1.** Extended version of table in *Figure 3c*, including all other olfactory-related neurons.

**Figure supplement 2.** Distribution of postsynaptic sites on the uPN dendrites.

---

In addition to inputs from Broad LNs (*Figure 4—figure supplement 2*), mPNs also receive up to 26% of inputs from 5 stereotypically connected, oligoglomerular LNs that we call Picky LNs (*Figure 4b, c*). While both Choosy LNs and Picky LNs are oligoglomerular and present distinct axons, the Choosy LNs are GABAergic whereas at least 4 of the 5 Picky LNs are instead glutamatergic (*Figure 2—figure supplement 1* for Picky LNs and *Figure 2* panels L-O in (*Thum et al., 2011*) for Choosy LNs; see also Supp. Fig. 2 of *Das et al. [2013]*). The difference in neurotransmitter is consistent with Picky LNs deriving from a different lineage than Choosy LNs (*Figure 1—figure supplement 3*). In addition, the two Choosy LNs present indistinguishable connectivity, whereas each Picky LN has its own preferred synaptic partners (see *Supplementary file 1* and *2*). Additionally, unlike the Choosy LNs, Picky LNs rarely target uPNs (*Figure 3—figure supplement 1*). Glutamate has been shown to act as a postsynaptic inhibitory neurotransmitter in the adult fly antennal lobe for both PNs and LNs (*Liu and Wilson, 2013*), and therefore in larva, Picky LNs may provide inhibition onto both mPNs and other LNs. Unlike the Broad LNs, which are panglomerular and axonless, the Picky LNs present separated dendrites and axons (*Figure 4b*). Collectively, Picky LN dendrites roughly tile the antennal lobe (*Figure 4b*). While some Picky LN axons target select uPNs, about 40% of Picky LN outputs are dedicated to mPNs or each other (*Figure 4c*; *Figure 2*). Similarly to the mPNs, Picky LNs 2, 3, and 4 receive inputs from unidentified non-ORN sensory neurons in the SEZ (*Figure 4b*, *Figure 2*).

Given that ORNs present overlapping odor tuning profiles (*Kreher et al., 2008*), we applied dimensionality-reduction techniques and discovered that ORNs cluster into 5 groups by odorant preference (*Figure 4—figure supplement 3*; see the Materials and methods section for more detail). This

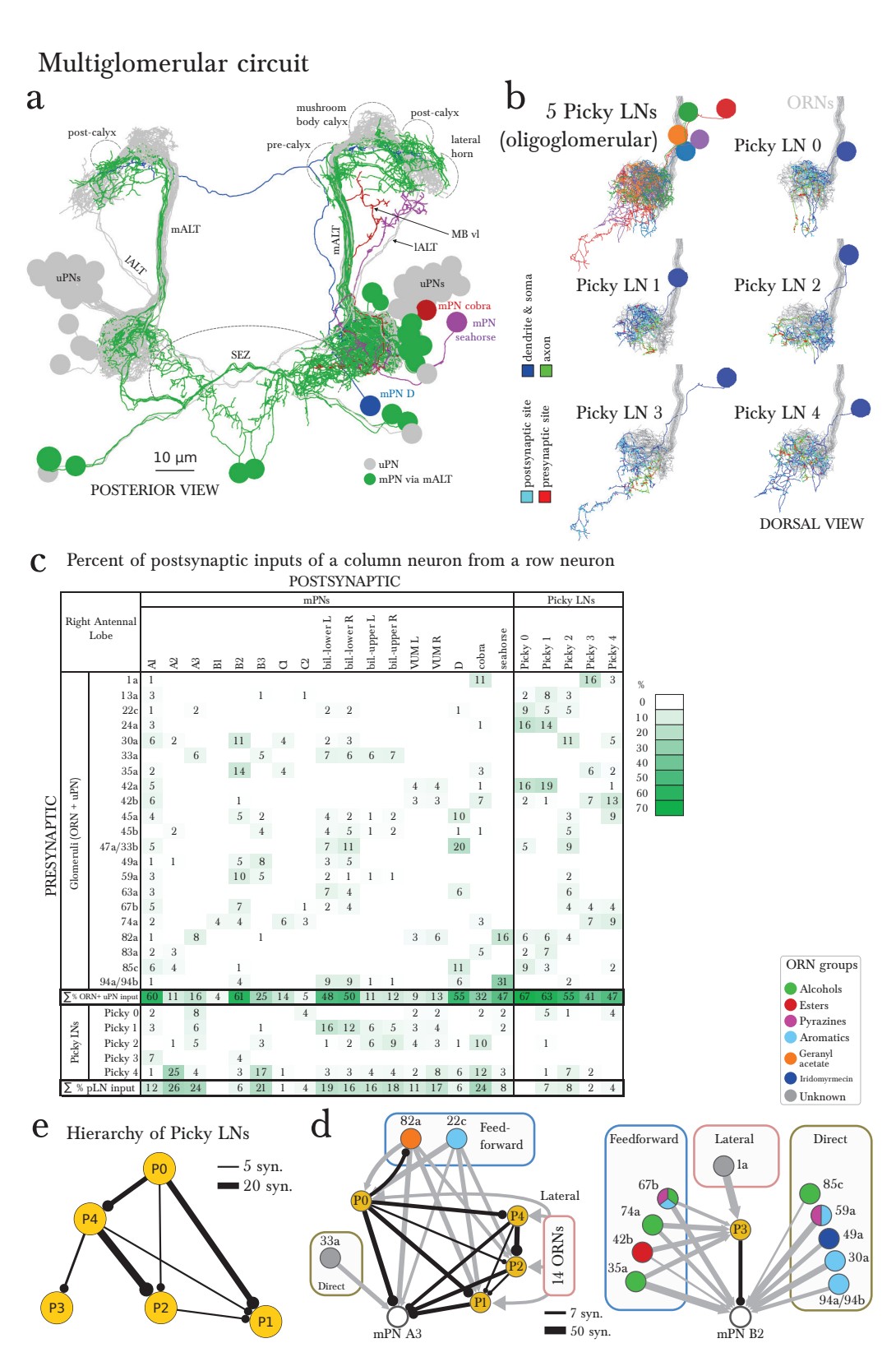

**Figure 4.** The multiglomerular circuit consists of 14 mPNs that project to the brain and 5 Picky LNs, each an identified neuron. (a) Posterior view of EM-reconstructed mPNs that innervate the right antennal lobe (in color; uPNs in grey for reference), each receiving inputs from a subset of olfactory

*Figure 4 continued on next page*

*Figure 4 continued*

glomeruli but many also from non-ORN sensory neurons in the subesophageal zone (SEZ). Most mPNs (green) project via the same tract as the uPNs (mALT). They can project via other tracts (other colors), but never via the mlALT used by the iPNs of the adult *Drosophila*. The mPNs project to many regions including a pre-calyx area, a post-calyx area, the lateral horn (LH) and the mushroom body vertical lobe (MB vl). mPNs are generated by diverse neuroblast lineages including BAlp4, BAla1, and others (*Das et al., 2013*). (**b**) Dorsal view of the EM-reconstructed Picky LNs shown together and individually. When shown individually, the Picky LNs are in 2 colors: blue for the dendrites and soma, and green for the axon. Zoom in to observe that presynaptic sites (red) are predominantly on the axon, whereas postsynaptic sites (cyan) are mostly on dendrites. Collectively, the dendritic arbors of the 5 Picky LNs tile the olfactory glomeruli. The dendrites of the Picky LN 3 and 4 extend significantly into the SEZ. They all originate from the same neuroblast lineage: BAla2 (*Das et al., 2013*). (**c**) Percentage of the total number of postsynaptic sites on the dendrite of a mPN or Picky LN (column neuron) that originate from a given glomerulus or Picky LN (row neurons). Here we define the glomerulus as connections from the ORN or via dendro-dendritic synapses from a given ORN's uPN. This is most relevant for mPN A1, which can receive more synapses from an ORN's uPN than the ORN itself (see suppl. Adjacency Matrix). We show the inputs to the mPNs and Picky LNs for the right antennal lobe, but for all bilateral mPNs (bil.-lower, bil.-upper, and VUM) we include inputs from both sides. We show only connections with at least two synapses, consistently found among homologous identified neurons in both the left and right antennal lobes. Percentages between 0 and 1 are rounded to 1, but totals are computed from raw numbers. Connections in this table are stereotyped (when comparing the left and right antennal lobes) and selective. Note that mPNs that receive many inputs from non-ORN sensory neurons in the SEZ have a low total of ORN+uPN input. For an extended version of this table that includes all LNs see *Figure 4—figure supplement 2*. (**d**) The direct upstream connectivity for two mPNs, with ORNs colored by the groups emerging from the PCA analysis of odor tuning. Connections from ORNs and Picky LNs to mPNs create 3 different types of motifs: *direct* excitatory connections from ORNs, *lateral* inhibitory connections from ORNs only via Picky LNs, and *feedforward loops* where an ORN connects both directly to the mPN and laterally through a Picky LN. Note that the activity of Picky LN 0 could alter the integration function for mPN A3 and indirectly for B2, as well as many other mPNs (not shown). Arrow thicknesses are weighted by the square root of the number of synapses between neurons. (**e**) The Picky LN hierarchy, dominated by Picky LN 0, here showing connections with 2 or more consistent synapses between bilaterally homologous neurons. Some of these connections are axo-axonic (see *Figure 4—figure supplement 3*).

The following figure supplements are available for figure 4:

**Figure supplement 1.** Six SEZ neurons receive specific inputs from some ORNs and from some antennal lobe LNs.

**Figure supplement 2.** Extended version of table in *Figure 4c*, including all other olfactory-related neurons.

**Figure supplement 3.** principal component analysis of odors leading to a principled clustering of orns.

**Figure supplement 4.** Pattern of ORN inputs onto Picky LNs.

helped interpret the pattern of ORNs onto Picky LNs and mPNs. We found that some Picky LNs aggregate similarly responding ORNs (*Figure 4d*; *Figure 4—figure supplement 4*). For example, Picky LN 2 receives inputs preferentially from ORNs that respond to aromatic compounds, and Picky LN 3 and 4 similarly for aliphatic compounds (esters and alcohols; *Figure 4—figure supplement 4*). On the other hand, Picky LN 0 and 1 aggregate inputs from ORNs from different clusters, suggesting that these Picky LNs may select for ORNs that are similar in a dimension other than odorant binding profile.

The stereotyped and unique convergence of different sets of ORNs onto both mPNs and Picky LNs, and the selective connections from Picky LNs to mPNs, suggest that each mPN responds to specific features in odor space, defined by the combinations of ORN and Picky LN inputs. These features are implemented through direct excitatory connections from ORNs or indirect inhibitory connections via Picky LNs (lateral inhibition; *Figure 4d*). Some ORNs affect the activity of the same mPN through both direct excitatory and lateral inhibitory connections through Picky LNs (incoherent feedforward loop, (*Alon, 2007*); *Figure 4d*). The combination of these motifs may enable an mPN to respond more narrowly to odor stimuli than the ORNs themselves, many of which are broadly tuned (*Kreher et al., 2008*), or to respond to a combinatorial function of multiple ORNs that describe an evolutionarily learned feature meaningful for the larva.

For example, one mPN (A1) reads out the total output of the uniglomerular system by integrating inputs across most ORNs and uPNs (*Figure 4c*). Another mPN (B2) could respond to the linear combination of ORNs sensitive to aromatic compounds (direct connections), but its response could change in the presence of alcohols and esters due to feedforward loops (*Figure 4d*). And mPNs A3 and B3 both collect inputs from ORNs (*Figure 4c,d*) known to respond to aversive compounds (22c, 45b, 49a, 59a, and 82a; [*Kreher et al., 2008*; *Ebrahim et al., 2015*]) or whose ORN drives negative chemotaxis (45a; [*Bellmann et al., 2010*; *Hernandez-Nunez et al., 2015*]). Additionally, mPN B3

receives inputs from 33a, an ORN whose receptor lacks a known binding compound, and therefore is likely narrowly tuned to other ecologically relevant odorants as was shown for 49a and the pheromone of a parasitic wasp (*Ebrahim et al., 2015*). The connectivity patterns of the 14 types of mPNs are vastly diverse from each other and likely each one extracts different features from odor space, often integrating inputs from non-ORN sensory neurons as well.

In contrast to the all-to-all connectivity of the Broad LNs, the Picky LNs synapse onto each other in a selective, hierarchical fashion (*Figure 4e*). The structure of the Picky LN hierarchy suggests that Picky LNs 0 and 3 can operate in parallel, while the activity of the other Picky LNs is dependent on Picky LN 0 (*Figure 4e*). These connections among Picky LNs include axo-axonic connections, and some Picky LNs receive stereotypic ORN inputs onto their axons (*Figure 4—figure supplement 4*). The stereotyped hierarchy among Picky LNs defines yet another layer of computations in the integration function of each mPN.

## Non-ORN sensory neurons and interactions among LNs could alter the operational state of the olfactory system

Picky LN 0 not only dominates the Picky LN hierarchy, and with it the multiglomerular system, but also may dramatically alter the inhibition of the entire olfactory system. This is because the main synaptic target of Picky LN 0 (*Figure 2*) is a bilateral, axonless, GABAergic LN called Keystone (*Figure 5a*; *Figure 2—figure supplement 1*), which in turn strongly synapses onto the Broad LN Trio–a major provider of presynaptic inhibition (*Figure 5b*). Interestingly, Keystone is also a major provider of presynaptic inhibition, but selectively avoids some glomeruli (*Figure 5c*; *Figure 3—figure supplement 1*). Therefore the wiring diagram predicts that these two parallel systems for presynaptic inhibition can directly and strongly inhibit each other (*Figure 5b*): homogeneous across all glomeruli when provided by the Broad LN Trio, and heterogeneous when provided by Keystone (*Figure 5c*). In conclusion, the circuit structure and the known synaptic signs predict that Picky LN 0 can promote a state of homogeneous presynaptic inhibition by disinhibiting the Broad LN Trio (*Figure 5d*).

The alternative state of heterogeneous presynaptic inhibition implemented by Keystone could be triggered by select non-ORN sensory neurons that synapse onto Keystone in the SEZ (*Figure 5a,b*). These non-ORN sensory neurons are the top inputs of Keystone and do not synapse onto any other olfactory LN. In contrast, ORNs that synapse onto Keystone also synapse onto the Broad LN Trio (*Figure 3—figure supplement 1*), suggesting a role for non-ORN sensory inputs in tilting the balance towards Keystone and therefore the heterogeneous state. However, the subset of ORNs that also synapse onto Picky LN 0 (*Figure 4c*) could oppose the effect of the non-ORN sensory neurons by inhibiting keystone and therefore disinhibiting the Broad LN Trio.

Neuromodulatory neurons could also affect the balance between Keystone and Broad LN Trio. Beyond the possible effect of volume release of serotonin (*Dacks et al., 2009*) and octopamine (*Linster and Smith, 1997*; *Selcho et al., 2012*) within the olfactory system, we found that these neuromodulatory neurons synapse directly and specifically onto Keystone or Broad LN Trio, respectively (*Figure 5b*). Beyond non-ORN inputs, ORNs synapse selectively onto these neuromodulatory neurons. Two ORNs (74a and 82a) synapse onto the serotonergic neuron CSD (*Roy et al., 2007*), and five ORNs (42b, 74a, 42a, 35a and 1a) onto an octopaminergic neuron (lAL-1; see Figure 4k in *Selcho et al., 2014*), suggesting that specific ORNs may contribute to tilting the balance between homogeneous and heterogeneous presynaptic inhibition via neuromodulation.

The only other provider of panglomerular presynaptic inhibition is the Broad Duet, which is the main provider of panglomerular postsynaptic inhibition. These neurons may operate similarly in both states given that they are inhibited by both Keystone and Broad LN Trio (*Figure 1c*). The higher fraction of inputs from Broad LN Trio onto Duet might be compensated by the fact that the Trio LNs inhibit each other (*Figures 3e*, *5b*), whereas the two Keystone LNs do not (*Figure 5b*). Therefore, potentially the Broad LN Duet could be similarly active in either state (*Figure 5d*).

## Some glomeruli are special-purpose

The possibility of heterogeneous presynaptic inhibition promoted by Keystone suggests that some ORNs can escape divisive normalization of their outputs relative to the rest. Not surprisingly, one such ORN is 49a (*Figure 5c*), which is extremely specific for the sexual pheromone of a parasitic wasp that predates upon larvae (*Ebrahim et al., 2015*). The other two ORNs that escape fully are 1a and 45b. 1a

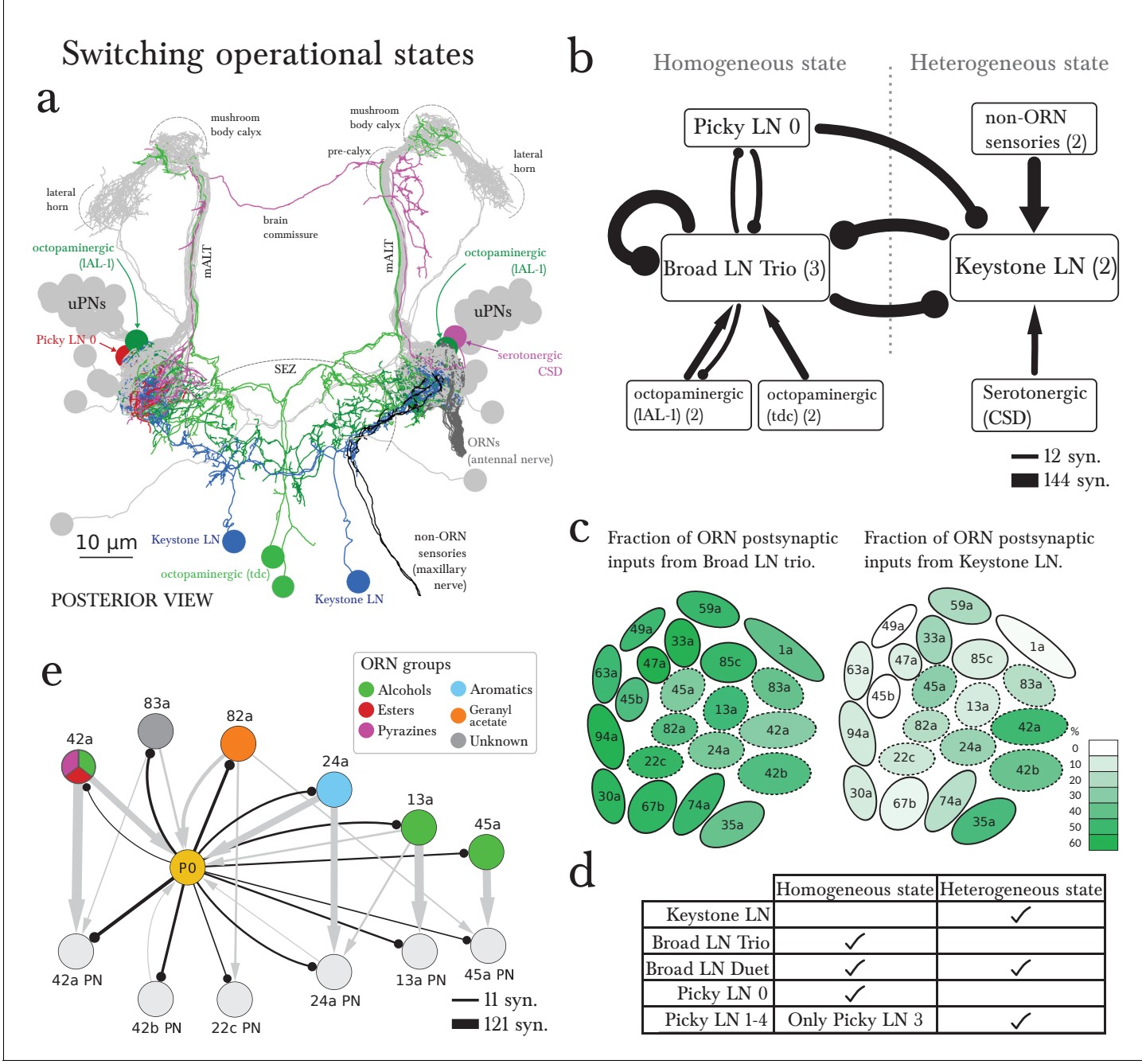

**Figure 5.** The wiring diagram suggests two operational states: homogeneous or heterogeneous presynaptic inhibition. (**a**) Posterior view of the EM-reconstructed neurons innervating the left antennal lobe that could govern the switch (uPNs in grey and right ORNs in dark grey for reference). The Keystone LN (blue) has a symmetric bilateral arbor and additionally innervates the SEZ, receiving inputs from non-ORN sensory neurons (in black). Neuromodulatory neurons that make direct morphological synapses onto LNs are serotonergic (CSD in pink; projects contralaterally after collecting inputs from near the MB calyx) and octopaminergic (lAL-1 and two tdc, in dark and light green), and all arborize well beyond the antennal lobe. Also included is Picky LN 0 (red). (**b**) A wiring diagram outlining the strong LN-LN connections, showing the core reciprocal inhibition between Broad LN Trio and Keystone that could mediate the switch between homogeneous (panglomerular) presynaptic inhibition and heterogenous (selective) presynaptic inhibition. For simplicity, neurons are grouped together if they belong to the same neuron type, with the number of neurons belonging to each group indicated in parentheses. Connections are weighted by the square root of the number of synapses between groups of neurons. The self-arrow for the Broad LN Trio represents the average number of synapses that one of the Trio neurons receives from the other two. Picky LN 0 inhibits Keystone, thereby disinhibiting the Broad LN Trio and promoting homogeneous presynaptic inhibition. The maxillary nerve sensory neurons are the top input providers of Keystone and may drive the system towards heterogeneous presynaptic inhibition (see C). The effect of direct inputs from neuromodulatory neurons is unknown, but at least it has been suggested that octopaminergic neurons may have an excitatory effect on inhibitory LNs

*Figure 5 continued on next page*

*Figure 5 continued*

(*Linster and Smith, 1997*). (c) Cartoon of glomeruli colored by the percentage of inputs onto ORN axon terminals provided by the Broad LN Trio and from Keystone, indicating the amount of presynaptic inhibition (onto ORNs) in either state. The inhibition provided by Broad LN Trio is much more uniform than the inhibition provided by Keystone. Dotted lines indicate glomeruli that receive Picky LN 0 input on either the ORN or uPN. (d) The LNs putatively active in each state. (e) Unlike other Picky LNs, Picky LN 0 makes synapses onto ORN axon terminals and many uPNs. Here connections with 2 or more synapses consistent between bilaterally homologous neuron pairs are shown. Arrow thicknesses are weighted by the square root of the number of synapses between neurons. With the exception of 45a, all shown ORNs and uPNs belong to glomeruli that synapse onto Picky LN 0 as well. Thus Picky LN 0 provides both pre- and postsynaptic inhibition to a small set of glomeruli.

The following figure supplement is available for figure 5:

**Figure supplement 1.** EM-reconstructed arbor of the descending neuron.

activation drives negative chemotaxis (Hernandez-Nuñez et al., unpublished). 45b senses compounds that elicit negative chemotaxis in larvae (*Kreher et al., 2008*). These three ORNs, and in particular 49a, are under strong postsynaptic inhibition by both Broad LN Duet and Choosy LNs (*Figure 3c*). In summary, reducing presynaptic inhibition in these 3 ORNs may enable larvae to perceive odors evolutionarily associated with life-threatening situations less dependently of the response intensity of other ORNs (i.e. overall odorant concentration). This is consistent with the finding that responses to aversive odors may rely on specific activity patterns in individual ORNs (*Gao et al., 2015*). The strong postsynaptic inhibition might be instrumental for their corresponding uPN to respond to the derivative of the ORN activity (with an incoherent feedforward loop; [*Alon, 2007*]), as shown in the adult fly (*Kim et al., 2015*), facilitating detection of concentration changes.

A key neuron in tilting the balance between homogeneous and heterogeneous presynaptic inhibition in the Broad-Keystone circuit is Picky LN 0 (*Figure 5b*). Remarkably, one of the two top ORN partners of Picky LN 0 is ORN 42a (*Figure 4c*), the strongest driver of appetitive chemotaxis in larvae (*Fishilevich et al., 2005*; *Asahina et al., 2009*; *Schulze et al., 2015*; *Hernandez-Nunez et al., 2015*). The connections of Picky LN 0 extend beyond that of other oligoglomerular LNs, and include both pre- and postsynaptic inhibition of a small subset of glomeruli, including 42a (*Figure 5e*). The wiring diagram therefore indicates that Picky LN 0, a likely glutamatergic LN, engages in seemingly contradictory circuit motifs: simultaneously inhibiting specific ORNs and their uPNs, while also disinhibiting them by inhibiting Keystone. The suppression of Keystone disinhibits the Broad LN Trio and therefore promotes homogeneous inhibition. However this is further nuanced by reciprocal connections between Picky LN 0 and Broad LN Trio (*Figure 5b*). This push-pull effect of glutamatergic LNs on PNs has been described for the olfactory system of the adult fly as conducive to more robust gain control and rapid transitions between network states (*Liu and Wilson, 2013*). This refined control could endow Picky LN 0-innervated glomeruli like 42a (*Figure 5e*) with the ability to better detect odor gradients, consistent with 42a being a strong and reliable driver of appetitive chemotaxis (*Fishilevich et al., 2005*; *Asahina et al., 2009*; *Schulze et al., 2015*).

Picky LN 0 and its push-pull effect on PNs not only can have an effect on positive chemotaxis but also on negative. A clear example is the 82a glomerulus (known to respond to an aversive odor that drives negative chemotaxis [*Kreher et al., 2008*]) which lacks a well-developed uPN but engages in strong connections with mPNs such as A3 (*Figure 4c,d*). We found that, like for the appetitive case of 42a, Picky LN 0 engages in both presynaptic inhibition onto 82a ORN and also postsynaptic inhibition onto mPN A3, one of the top PNs of 82a ORN. And like other ORNs mediating aversive responses (e.g 49a), the 82a uPN is also under strong postsynaptic inhibition (*Figure 3c*).

Finally, we found evidence that an individual glomerulus can have a global effect on the olfactory system. All LNs (except Picky LN 3) receive inputs from Ventral LN (*Figure 1c*,*Figure 3—figure supplement 1*), an interneuron of unknown neurotransmitter, which is primarily driven by the 13a glomerulus. This suggest that 13a, an ORN sensitive to alcohols (*Kreher et al., 2008*), could potentially alter the overall olfactory processing.

## Feedback from the brain

In the mammalian olfactory bulb, descending inputs from the brain target granule cells (the multiglomerular inhibitory LNs), shaping the level of inhibition (*Balu et al., 2007*). In addition to descending

neuromodulatory neurons (CSD; *Figure 5a*), in the larva we found a descending neuron (*Figure 5—figure supplement 1*) that targets specific mPNs and LNs (*Figure 4—figure supplement 2*). In addition to other mPNs, this descending neuron targets the two mPNs that we postulate are aversive (mPNs A3 and B3). Together with the axo-axonic inputs it receives from 45a ORN (an aversive ORN, [*Bellmann et al., 2010*; *Hernandez-Nunez et al., 2015*]), this descending neuron is associated with the processing of aversive stimuli. Additional descending neurons affecting PNs and LNs might exist but were beyond the scope of this study, where we focused on neurons directly synapsing with ORNs.

## Discussion

The glomerular olfactory system of the larva develops in a similar fashion to the vertebrate olfactory bulb where the afferents (i.e. ORNs) organize the central neurons, unlike in the adult fly (*Prieto-Godino et al., 2012*). In zebrafish, GABAergic LNs provide depolarizing currents to PNs (mitral cells) via gap junctions at low stimulus intensities, enhancing low signals, and inhibit the same PNs at high stimulus intensity via GABA release, implementing a form of gain control (*Zhu et al., 2013*). This role is played by a class of panglomerular excitatory LNs in the adult fly that make gap junctions onto PNs and excite inhibitory LNs (*Yaksi and Wilson, 2010*). In the larva, all panglomerular neurons are GABAergic; if any were to present gap junctions with uPNs, a cell type for gain control in larva would be equivalent to the one in zebrafish. Particularly good candidates are the Broad LN Duet, which provide the bulk of feedforward inhibitory synapses onto uPNs in larva. Interestingly, postsynaptic inhibition might not be mediated by GABA in the adult fly (supp. fig. 5 in *Olsen and Wilson [2008]*), rendering olfactory circuits in larva more similar to vertebrates. Presynaptic inhibition exists both in the adult fly and, as suggested by the present work, in larva, and is mediated by the same kind of panglomerular GABAergic neurons (the Broad Trio LNs in larva; and see [*Olsen and Wilson, 2008*]).

The uniglomerular circuit is the most studied in all species both anatomically and physiologically. We found that each uPN receives an unusually large number of inputs from an individual ORN compared to other sensory systems in the larva (*Ohyama et al., 2015*). This large number of morphological synapses could be interpreted as a strong connection, which would support faster or more reliable signal transmission. In the adult fly, the convergence of multiple ORNs onto an individual PN enables both a fast and reliable PN response to odors (*Bhandawat et al., 2007*). The temporal dynamics of crawling are far slower than that of flying, and therefore we speculate that the integration over time of the output of a single ORN might suffice for reliability, demanding only numerous synapses to avoid saturation.

Positive, appetitive chemotaxis involves odor gradient navigation, leading to a goal area where food is abundant which may overwhelm olfaction. We postulate that navigation and feeding correspond to the homogeneous and heterogeneous states of presynaptic inhibition that we described. During navigation, homogeneous presynaptic inhibition (via Broad LN Trio) could best enhance salient stimuli and therefore chemotaxis, enabling the olfactory system to operate over a wide range of odorant intensities (*Asahina et al., 2009*). During feeding, strongly stimulated ORNs could scale down the inputs provided by other, less stimulated, ORNs. In other words, if homogeneous presynaptic inhibition persisted during feeding, the larvae would lose the ability to detect important odorants that are likely to be faint, for example the scent of a predator such as a parasitic wasp via 49a (*Ebrahim et al., 2015*). The larva can selectively release presynaptic inhibition via Keystone, which provides presynaptic inhibition to appetitive glomeruli while also inhibiting the Broad LN Trio–the major providers of panglomerular presynaptic inhibition. So the larva could feed and remain vigilant to evolutionarily important cues at the same time. Not surprisingly, the switch might be triggered by neuromodulatory neurons and non-ORN sensory neurons, potentially gustatory, that synapse onto Keystone.

In addition to the uniglomerular system that is present across multiple vertebrate and invertebrate species (*Satou, 1992*; *Wang et al., 1998*; *Vosshall et al., 2000*), we found, in the *Drosophila* larva, a multiglomerular system that presumably performs diverse processing tasks already at the first synapse. One such task could be the detection of concentration gradients for some odorant mixtures, suggesting an explanation for the observation that some ORNs can only drive chemotaxis when co-activated with other ORNs (*Fishilevich et al., 2005*). Similar glomerular-mixing circuits have been described in higher brain areas (lateral horn) of the fly (*Wong et al., 2002*; *Fişek and Wilson, 2014*) and of mammals (*Sosulski et al., 2011*). We hypothesize that in the larva, the morphological adaptations to a life of burrowing might have led to specific adaptations, relevant to an animal that

eats with its head, and therefore the dorsal organ housing the ORNs, immersed in food. It is perhaps not surprising that we found multisensory integration across ORNs and non-ORNs (likely gustatory) already at the first synapse. And we hypothesize that the pooling of chemosensors (ORNs and non-ORNs) onto mPNs and Picky LNs may be related to the reduction in the number of ORNs relative to insects with airborne antennae.

With our complete wiring diagram of this tractable, transparent model system and genetic tools for manipulating and monitoring the activity of single identified neurons, we have now the opportunity to bridge the gap between neural circuits and behavior (*Carandini, 2012*).

## Materials and methods

### Electron microscopy and circuit reconstruction

We reconstructed neurons and annotated synapses in a single, complete central nervous system from a 6-h-old [*iso*] *Canton S G1 x w*$^{1118}$ larva imaged at 4.4 x 4.4 x 50 nm resolution, as described in *Ohyama et al. (2015)*. The volume is available at http://openconnecto.me/catmaid/, titled "'acar-dona_0111_8'. To map the wiring diagram we used the web-based software CATMAID (*Saalfeld et al., 2009*), updated with the novel suite of neuron skeletonization and analysis tools (*Schneider-Mizell et al., 2016*), and applied the iterative reconstruction method described in *Ohyama et al. (2015)*; *Schneider-Mizell et al. (2016)*.

### Immunolabeling and light microscopy

CNS was dissected from 3rd instar larvae. 4% formaldehyde was used as fixative for all antibodies except anti-dVGlut that required bouin fixation (*Drobysheva et al., 2008*). After fixation, brain samples were stained with rat anti-flag (1:600, Novus Biologics) and chicken anti-HA (1:500, Abcam, ab9111) for labeling individual neurons in the multi-color flip-out system (*Nern et al., 2015*), while mouse anti-Chat (1:150, Developmental Studies Hybridoma Bank, ChaT4B1) and rabbit anti-GABA (1:500, Sigma A2052, Lot# 103M4793) or anti-dVGlut (*Daniels et al., 2004*) were used for identifying neurotransmitters. Antibodies were incubated at 4°C for 24 hr. Preparations were then washed 3 times for 30 min. each. with 1% PBT and then incubated with secondary antibodies (including: goat anti-Mouse Alexa Fluor 488, goat anti-rabbit Alexa Fluor 568, donkey anti-rat Alexa Fluor 647, Thermofisher; and goat anti-chicken Alexa Fluor 405, Abcam) at 1/500 dilution for 2 hr at room temperature, followed by further washes. Nervous systems were mounted in Vectashield (Vector Labs, Burlingame, CA) and imaged with a laser-scanning confocal microscope (Zeiss LSM 710).

### Clustering of ORNs by PCA of their responses to odors

Extensive screens have been conducted to identify which odorants activate each ORN (*Kreher et al., 2008*; *Montague et al., 2011*; *Mathew et al., 2013*). These data can be used as a starting point to determine whether the multiglomerular circuit extracts relevant components of odorant physical descriptor space, that is, the chemical structure of the odorant as sampled by ORNs. Using the data from (*Kreher et al., 2008*) and (*Montague et al., 2011*), we conducted a dimensionality reduction via PCA followed by a clustering analysis, and then used the data from (*Mathew et al., 2013*) to verify our findings. To determine how ORNs encode odors we followed the PCA analysis in ORN space performed in (*Haddad et al., 2010*) adding the data of pyrazines from (*Montague et al., 2011*). Then, using the Scree test we selected the first 3 components of the PCA as the relevant ones to use for clustering, and we ran the clustering minimization using the affinity propagation algorithm (which doesn't require the number of clusters as an input) (*Figure 4—figure supplement 3b,c*). Four of the obtained clusters correspond very well with odorant type (alcohols, aromatics, esters, and pyrazines; *Figure 4—figure supplement 3d*). The fifth cluster is mixed and mainly includes odorants with very low or no ORN response. Consistent with that, k--means clustering in the 32-dimensional odorant physical descriptor space described in (*Haddad et al., 2008*) results in 5 clusters, 4 of them matching the non-mixed clusters obtained in ORN space (*Figure 4—figure supplement 3a*).

To determine which ORNs encode the regions of each cluster, we projected back the centroid of each cluster onto ORN space using the inverse transformation (*Figure 4—figure supplement 3e*). Different subsets of ORNs were more likely to encode each cluster. The projections of the cluster

centroids in ORN space are not discrete numbers; in order to make these results easier to interpret a threshold can be established to determine which ORNs encode a cluster centroid and which ones don't. A suitable approach is to use Otsu's method, which can be considered a one-dimensional discrete analog of Fisher's discriminant analysis (*Otsu, 1975*). We obtained a threshold of 0.4725, which we used to determine the ORNs that encode each cluster (dashed red line in *Figure 4—figure supplement 3e*).

In (*Mathew et al., 2013*) a set of odorants that specifically activate single ORNs at low concentrations were identified. These data can easily be used to cross-validate the predicted receptive field of the different ORNs in our analysis. Four of the odorants tested were alcohols and activated Or13a, 35a, 67b and 85c all of which are in our alcohols group. Other three were aromatics and activated Or22c, 24a and 30a, which are all in our aromatics group. One was a pyrazine and activated Or33b which is in our pyrazine group. Finally pentyl acetate (an ester) activated 47a which is in our esters group. The other odorants in (*Mathew et al., 2013*) were in regions of odor space (mostly ketones and aldehydes) that were not sampled in (*Kreher et al., 2008*) or (*Montague et al., 2011*) and therefore their responses cannot be predicted with our analysis. As more datasets are collected, approaches like the one we present here can be used to better establish the receptive field of each ORN.

## Acknowledgements

We thank Ingrid Andrade, Anton Miroschnikow, Ilona Brueckmann, Ivan Larderet, Volker Hartenstein, Bruno Afonso and Philipp Schlegel for contributing 15.5% of all reconstructed arbor cable, and Gaetan Vignoud for assistance with the PCA. We thank Rachel Wilson, Kathy Nagel, Andreas Thum, Bertram Gerber, Markus Knaden, Marc Gershow, Mei Zhen and Ibrahim Tastekin for constructive comments and discussions. ADTS thanks "The Harvard Brain Initiative Collaborative Seed Grant Program", the NIH Pioneer Award (8DP1GM105383), NIH PO1 (1P01GM103770) and NSF BRAIN Initiative (NSF-IOS-1556388). MEB thanks the NSF Physics of Living Systems Student Network. ML and AK acknowledge support of the Spanish Ministry of Economy and Competitiveness, 'Centro de Excelencia Severo Ochoa 2013-2017', SEV-2012-0208 and grants MICINN BFU2011-26208. AK acknowledges the support of the 'laCaixa' International PhD programme. We thank the Fly EM Project Team at HHMI Janelia for the gift of the EM volume, the HHMI visa office, and HHMI Janelia for funding.

## Additional information

### Funding

| Funder | Grant reference number | Author |
|---|---|---|
| National Institutes of Health | NIH Pioneer Award, 8DP1GM105383 | Matthew E Berck<br>Lindsey Claus<br>Luis Hernandez-Nunez<br>Guangwei Si<br>Christopher J Tabone<br>Aravinthan DT Samuel |
| National Institutes of Health | PO1, 1P01GM103770 | Matthew E Berck<br>Lindsey Claus<br>Luis Hernandez-Nunez<br>Guangwei Si<br>Christopher J Tabone<br>Aravinthan DT Samuel |
| Howard Hughes Medical Institute | | Feng Li<br>James W Truman<br>Rick D Fetter<br>Albert Cardona |
| National Science Foundation | Brain Initiative, NSF- IOS-1556388 | Matthew E Berck<br>Lindsey Claus<br>Luis Hernandez-Nunez<br>Guangwei Si<br>Christopher J Tabone<br>Aravinthan DT Samuel |

| Harvard University | Collaborative Seed Grant | Matthew E Berck<br>Lindsey Claus<br>Luis Hernandez-Nunez<br>Guangwei Si<br>Christopher J Tabone<br>Aravinthan DT Samuel |
| --- | --- | --- |
| National Science Foundation | Physics of Living Systems Student Network | Avinash Khandelwal<br>Matthieu Louis |
| Spanish Ministry of Economy and Competitiveness | MICINN BFU2011-26208 | Avinash Khandelwal<br>Matthieu Louis |
| Centro de Excelencia Severo Ochoa | | Avinash Khandelwal<br>Matthieu Louis |
| Ministerio de Ciencia e Innovación | BFU2011-26208 | Avinash Khandelwal<br>Matthieu Louis |

The funders had no role in study design, data collection and interpretation, or the decision to submit the work for publication.

## Author contributions
MEB, AK, ADTS, AC, Conception and design, Acquisition of data, Analysis and interpretation of data, Drafting or revising the article; LC, CJT, Acquisition of data, Analysis and interpretation of data; LH-N, GS, Analysis and interpretation of data, Drafting or revising the article; FL, JWT, Acquisition of data, Analysis and interpretation of data, Drafting or revising the article; RDF, Acquisition of data, Drafting or revising the article; ML, Conception and design, Analysis and interpretation of data, Drafting or revising the article

## Author ORCIDs
Christopher J Tabone, http://orcid.org/0000-0001-8746-0680
Matthieu Louis, http://orcid.org/0000-0002-2267-0262
Albert Cardona, http://orcid.org/0000-0003-4941-6536

# Additional files

## Supplementary files
• Supplementary file 1. Adjacency matrices with the complete synaptic connectivity of the wiring diagram of the left antennal lobe.

• Supplementary file 2. Adjacency matrices with the complete synaptic connectivity of the wiring diagram of the right antennal lobe.

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
