## [Decision Letter]

Thank you for submitting your work entitled "The wiring diagram of a glomerular olfactory system" for consideration by *eLife*. Your article has been favorably evaluated by K VijayRaghavan (Senior editor) and three reviewers, one of whom, Ronald L. Calabrese, is a member of our Board of Reviewing Editors.

The reviewers have discussed the reviews with one another and the Reviewing Editor has drafted this decision to help you prepare a revised submission.

Summary:

The authors present an em-level reconstruction of the complete wiring diagram of the left and right antennal lobes of *Drosophila* larva. This feat in important genetic model organisms is a first and is also a necessary step in the detailed experimental and theoretical analyses of circuit function bridging the gap between circuits and behavior. This work is informed by a multitude of physiological studies (albeit mainly in adults) of olfactory lobe function and analyses of behavior especially chemotaxis in the larva. The beauty and elegance of the work is hard to overstate, and its impact will be felt far into the future especially in the field of olfaction.

The work builds on methods previously published in *eLife* by the Cardona group and those methods are consequently only cursorily described in this manuscript and no primary em data is shown. Nevertheless, the figures presented are detailed and convincing and supported by ample supplemental data (these could be better integrated into the main text). Quantification of synaptic contacts is abundant and presented in easy to read tables, and enhanced by intuitive 'functional' circuit diagrams. Writing is succinct and clear.

The work opens the *Drosophila* larval model for further analyses of chemotaxis at the cellular, network, and behavioral level and complements studies in more complex vertebrate systems where identified neurons are less defined or accessible and genetic techniques less developed. The paper as presented is compelling and should be of interest to a wide range of neuroscientists, especially those studying olfaction and orientation/taxes in stimulus gradients.

Essential revisions:

The reviewers were all in agreement about the elegance and impact of the work, but were concerned about the blurring of the lines of the structure (elegantly explicated) and the *putative* function. The manuscript should be revised to more clearly delineate the boundary. The raw comments of the reviewers are provided to guide the revision process.

*Reviewer #1:*

1) While the writing is very clear, the Results section is difficult to read because of the tremendous amount of data presented and complexity of the functional arguments. This is not helped by the picky and choosy designations for LNs; please give more functional names to these types – trio and duo are hard enough to keep straight. Do everything possible to refer to the figures directly when making functional arguments.

2) While the complete wiring diagram of such a complex network is a tremendous feat for which the authors are to be congratulated and it is a necessary step for understanding network function, it is not sufficient for understanding network function, and the authors should be more circumspect in making functional claims, especially in the Abstract.

*Reviewer #2:*

1) Abstract: What exactly is meant by the term "internalized valences"?

2) Abstract: The authors mention that a wiring diagram with synaptic resolution is "unknown for any glomerular olfactory system". A synaptic connectivity diagram has been published for several glomeruli in the adult fly by Rybak et al. (J Comp Neurol. 2016 Jan 18. doi: 10.1002/cne.23966). Since this work has been published only recently, it might have escaped the authors' attention. But this publication should definitely be cited and mentioned in detail. The strength of the present manuscript relies on the completeness of the larval antennal lobe, and this could be pointed out instead.

3) Abstract: The authors refer to the mushroom body as "the learning and memory center" and to the lateral horn as the "center for innate behaviors". This is certainly oversimplified, and actually wrong. The mushroom body is, at least in adult flies, involved in innate behaviors, and there is no evidence for the lateral horn not to play a role in learning and memory.

4) Abstract: The authors draw the conclusion from the synaptic wiring between local neurons that multiglomerular projection neurons "extract complex features from odor space". First, from anatomical data one can certainly formulate hypotheses about the physiological function of the network. However, such hypotheses have to be tested physiologically before a firm conclusion can be drawn. Second, it remains unclear which 'complex features' the authors refer to, and how they are 'extracted'.

5) Introduction: The authors write that projection neurons targeting two brain areas are common between vertebrates and invertebrates. I suggest writing "mammals and insects", because there are too many invertebrates in which this is not the case, and I am not sure whether it is true for all vertebrates.

6) The authors state that one brain area is associated with learning and memory, and the other with innate behavior. For the insect mushroom body and the lateral horn, see comment 3 above. For the amygdala and the piriform cortex, this is also not true. The amygdala is well known to be involved in certain types of associative learning, and the piriform cortex is likely to be involved in innate behavior as well.

7) Introduction: The authors mention that a wiring diagram with synaptic resolution is "unknown for any glomerular olfactory system". See comment 2 above.

8) Introduction: The authors should mention in the Introduction that in *Drosophila* there are first, second and third instar larvae. Most behavioral studies have been done in third instar larvae, whereas this study refers to a dataset obtained from a first instar larva. It should be mentioned if the olfactory systems of these different instars differ.

9) Introduction: The authors mention that ORN responses "represent both the first derivative of the odor concentration and the intensity of the odor". First, ORNs do not only encode the first derivate of odor concentration (i.e., its change over time), but also its actual time course. Second, I do not understand the difference between odor concentration and odor intensity. Third, one has to carefully differentiate between 'odorant' (i.e., the chemical compound) and 'odor' (i.e., the neuronal representation of the stimulus evoked by the odorant).

10) Introduction: The authors correctly and cautiously state: "the LNs structure a circuit that *putatively* implements a bistable inhibitory system". Then they continue that "one state computes odor saliency". What exactly does saliency mean? Do the authors refer to the ability of the stimulus to provoke a behavioral response? If so, this statement should also be toned down as it is a hypothetical assumption. The same applies to the second state. Please separate more clearly your anatomical findings from potential functions of the circuitry.

11) Results: Please do not refer to the mushroom body as a center for learning and memory in contrast to the lateral horn (see comment 3).

12) Results: The authors state that duet LNs play a "far stronger role in postsynaptic inhibition". Actually, the result is that those duet neurons have more synapses onto PNs than the trio LNs. All conclusions beyond that are hypothetical, i.e., how strong the inhibition of the postsynaptic neuron actually is.

13) Results: The subheading "Extracting features from odor space at the first synapse" refers to too speculative interpretation of the anatomical data.

14) Results: The terms 'Choosy LNs' and 'Picky LNs' I found somehow confusing. Choosy LNs are GABergic, and Picky LNs mostly glutamatergic, correct? Both names indicate that they selectively innervate only few glomeruli. Maybe one can give them names that are more indicative of their differential connectivity.

15) Results: The authors state that "Picky LNs may select for ORNs that are similar in a dimension other than odor preference". I assume the authors mean the array of odorants binding to the receptors ('odorant selectivity' or "odorant binding profile"), because with "odor preference" I intuitively think of the animals' behavior.

16) Results: The authors often use the term 'feature', and it is not really clear what exactly that means. Perhaps one could replace that term with "combinations of ORN inputs".

17) Results: The subheading "Non-ORN sensory neurons and interactions among LNs can alter the operational state of the olfactory system" is too speculative.

Operational states the authors have not investigated here. And in the entire Results section, I suggest that the authors make clear what they have analyzed (anatomical connectivity) and what is a potential interpretation (functional consequences from the circuitry, e.g., which neuron might provide stronger or weaker inhibition).

18) Discussion: This section is perfectly clear.

19) The reader might wonder why in this anatomical study EM pictures are not included. I suggest that an indication that this work is based on a published EM stack, and that the volume used here is available online, could appear already early in the manuscript. The authors indicate that only in the Materials and methods section.

Reviewer #3:

In Figure 1—figure supplement 2 ORN synapses are rendered in the same color as the skeleton. This is not very helpful, since it is not clear what is a synapse and what is part of the arbor.

"Therefore Broad LNs may keep the entire olfactory system within the dynamic range to remain responsive to changes in odor intensities." Consider rephrasing. It is not that these neurons keep the system within the dynamic range, but rather these properties define the dynamic range. Same later in the text, when the authors state: "to maintain uPN output within the dynamic range".

"Importantly, Broad LNs also synapse onto each other (Figure 2C, E) like in the adult (Okada et al., 2009), suggestive of a mechanism for sequential recruitment as overall stimulus intensity increases". It is not clear how this would work given that LNs are inhibitory. Please explain.

Figures 2 and 3: order of panel E and D is switched.

It is hard to understand from the text how Choosy LNs that are driven by only a small subset of glomeruli can drive postsynaptic inhibition of most other glomeruli. Inspecting Figure 3—figure supplement 1 reveals that Choosy LNs receive ORN input in 9 glomeruli, but synapse on uPNs in 17 glomeruli. This should be explained more clearly.

---

## [Author Response]

Essential revisions:

The reviewers were all in agreement about the elegance and impact of the work, but were concerned about the blurring of the lines of the structure (elegantly explicated) and the putative function. The manuscript should be revised to more clearly delineate the boundary. The raw comments of the reviewers are provided to guide the revision process.

Reviewer #1:

1) While the writing is very clear, the Results section is difficult to read because of the tremendous amount of data presented and complexity of the functional arguments. This is not helped by the picky and choosy designations for LNs; please give more functional names to these types – trio and duo are hard enough to keep straight. Do everything possible to refer to the figures directly when making functional arguments.

2) While the complete wiring diagram of such a complex network is a tremendous feat for which the authors are to be congratulated and it is a necessary step for understanding network function, it is not sufficient for understanding network function, and the authors should be more circumspect in making functional claims, especially in the Abstract.

We agree with the reviewer about the need to clearly separate the wiring diagram from our interpretation of it. We took an approach that intermingled the description of particular novel aspects of the wiring diagram of the antennal lobe with their interpretation, with the goal of facilitating understanding of each module separately, by providing both the context from the literature, which is rich in behavioral and functional data, and our synthesis that led us to formulate specific hypotheses about circuit functions. Writing in this way enabled our many informal reviewers prior to submission to *eLife* to better follow the results, building up one piece at a time. We have realized that this approach has led to specific instances where the line was blurred between data and interpretation, and we thank the reviewer for identifying these. We have expanded the text and clarified these instances as is appropriate.

To address this, we reworded multiple sections of the paper to clarify that functional conclusions are derived from the synaptic connectivity, neurotransmitter signatures, circuit structure, and prior published work. We followed the same strategy to combine our novel circuit data with the published data, with the goal of proposing a set of hypotheses for future experiments.

We understand the overwhelmingness of the number of neurons and therefore the issues that arise with naming them, for which we chose the way that proved least confusing even to ourselves. In the case of Picky and Choosy LNs, which was already raised by our colleague Leslie Vosshall in a public review in biorxiv, we understand that the names are rather interchangeable. But this is on purpose: these two neuron types, while deriving from different neurons lineages and presenting different neurotransmitters (which makes them deserving of different names), their connectivity is similar in that they both integrate inputs from a selective subset of ORNs and synapse onto a subset of PNs. To address the potential confusion, we have introduced a number of changes. First, we have changed a previously supplemental figure into a main figure (the revised manuscript’s Figure 2). This figure clearly shows the differences in the synaptic partners for the picky and choosy LNs. Additionally, we added an extra figure supplement to Figure 1 (Figure 1—figure supplement 5) that illustrates the different lineages of all 14 of the LNs. The figure makes it clear that Broad LNs, Choosy LNs, Picky LNs, Ventral LN, and Keystone belong to different developmental lineages. Additionally, we have added an extra paragraph describing the differences between LN types before we discuss their functional roles in the circuits of the antennal lobe. Finally, we have added additional language when reintroducing the Picky and Choosy LNs that highlights the differences between the 2 LN types (glutamate vs. GABA; targeting very few uPNs vs. targeting most uPNs; different developmental lineages). We hope that this will clarify the different characteristics of these two cell types and justifies their names.

Reviewer #2:

1) Abstract: What exactly is meant by the term "internalized valences"?

When we say “internalized valences,” we meant both learned valences (via associative learning) and innate valences (learned via evolution), such as appetitive or aversive. We replaced “internalized” with “learned and innate”, to make it more explicit. The use of “valence” is standard in the field, see for example:

Anderson, Adam K., et al. "Dissociated neural representations of intensity and valence in human olfaction." Nature neuroscience 6.2 (2003): 196-202.

Zelano, C., et al. "Dissociated representations of irritation and valence in human primary olfactory cortex." Journal of neurophysiology97.3 (2007): 1969-1976.

2) Abstract: The authors mention that a wiring diagram with synaptic resolution is "unknown for any glomerular olfactory system". A synaptic connectivity diagram has been published for several glomeruli in the adult fly by Rybak et al. (J Comp Neurol. 2016 Jan 18. doi: 10.1002/cne.23966). Since this work has been published only recently, it might have escaped the authors' attention. But this publication should definitely be cited and mentioned in detail. The strength of the present manuscript relies on the completeness of the larval antennal lobe, and this could be pointed out instead.

(See also related points 7 and 11) Thank you for bringing this very recent paper to our attention. We now have read the Rybak et al. paper and incorporated its findings into our interpretation of the wiring diagram. In the Introduction, we state that while the connectivity of a few glomeruli has been partially reconstructed in the adult fly, what we are presenting is the complete number and morphology of cell types as well as the complete circuit structure with synaptic resolution, along with neurotransmitters of the LNs. Since LNs in the Rybak paper are not fully reconstructed, interglomerular interactions (which is a major aspect of our paper) are only very partially addressed. For this reason, we cite Rybak two more times when talking about general intraglomerular LN-PN connections in the uniglomerular circuit: both when discussing connections between LNs and PNs and proximal vs distal connections of ORNs and LNs onto PNs (uPNs).

3) Abstract: The authors refer to the mushroom body as "the learning and memory center" and to the lateral horn as the "center for innate behaviors". This is certainly oversimplified, and actually wrong. The mushroom body is, at least in adult flies, involved in innate behaviors, and there is no evidence for the lateral horn not to play a role in learning and memory.

(See also point 6) We have changed the language of the paper when referring to the mushroom body and lateral horn. To clarify that these roles are not the exclusive roles of these brain regions, we now say that the mushroom body is “required” for learning and memory (based on de Belle & Heisenberg, 1994), and that the lateral horn is “implicated in” or “mediates some” innate behaviors (as shown in numerous papers where removing the mushroom body does not remove the subset of innate behaviors under study). Furthermore, to indicate that what motivated our simplification – with the understanding that our wording was a simplification – is that many leaders in the field use this shorthand way of referring to the MB and LH in papers and talks.

4) Abstract: The authors draw the conclusion from the synaptic wiring between local neurons that multiglomerular projection neurons "extract complex features from odor space". First, from anatomical data one can certainly formulate hypotheses about the physiological function of the network. However, such hypotheses have to be tested physiologically before a firm conclusion can be drawn. Second, it remains unclear which "complex features" the authors refer to, and how they are "extracted".

Generally, we have changed the language when referring to functional roles of the circuitry to better clarify that we are hypothesizing based on synaptic connectivity and circuit structure. When we refer to “features from odor space,” we are talking about the pattern where a set of ORNs connect to a specific mPN and Picky LNs within the multiglomerular circuit, and how the collective activity of these ORNs can putatively represent a specific olfactory object (in odor space) such as a particular food, a predator, etc. We discuss specific features when talking about the putative functions of certain mPNs, such as: overall odorant concentration, odorant type (aromatic vs. aliphatic), and valence (clustering ORNs that respond to aversive compounds or ORNs that produce aversive behavior when optogenetically activated). The “extraction” is the interesting task here: we found that this could take place at the first synapse, because of the presence of multiglomerular neurons that might combine information in sophisticated ways involving lateral interactions between glomeruli mediated by inhibitory LNs. We have completely rewritten the Abstract to comply with space limits and took the opportunity to more carefully focus the emphasis on the circuit motifs rather than our interpretation of them.

5) Introduction: The authors write that projection neurons targeting two brain areas are common between vertebrates and invertebrates. I suggest writing "mammals and insects", because there are too many invertebrates in which this is not the case, and I am not sure whether it is true for all vertebrates.

We consider that most non-parasitic invertebrates follow this pattern, and that all vertebrates (not chordates) follow this pattern as well. In any case we have no axe to grind here and have changed the manuscript to say “mammals and insects”, which are the groups to which the species cited in the manuscript belong.

6) The authors state that one brain area is associated with learning and memory, and the other with innate behavior. For the insect mushroom body and the lateral horn, see comment 3 above. For the amygdala and the piriform cortex, this is also not true. The amygdala is well known to be involved in certain types of associative learning, and the piriform cortex is likely to be involved in innate behavior as well.

See related comment to point 3 above.

7) Introduction: The authors mention that a wiring diagram with synaptic resolution is "unknown for any glomerular olfactory system". See comment 2 above.

See comment 2 above.

8) Introduction: The authors should mention in the Introduction that in Drosophila there are first, second and third instar larvae. Most behavioral studies have been done in third instar larvae, whereas this study refers to a dataset obtained from a first instar larva. It should be mentioned if the olfactory systems of these different instars differ.

We now mention that most behavioral experiments are done in 2nd and 3rd instar larvae and that the EM volume is of a 1st instar larvae.

9) Introduction: The authors mention that ORN responses "represent both the first derivative of the odor concentration and the intensity of the odor". First, ORNs do not only encode the first derivate of odor concentration (i.e., its change over time), but also its actual time course. Second, I do not understand the difference between odor concentration and odor intensity. Third, one has to carefully differentiate between "odorant" (i.e., the chemical compound) and "odor" (i.e., the neuronal representation of the stimulus evoked by the odorant).

First, thanks for pointing out the confusion between odor and odorant. Indeed, the dictionary is clear on these, yet the literature not so much. At least one paper in *C. elegans* (Albrecht, Dirk R., and Cornelia I. Bargmann. "High-content behavioral analysis of *Caenorhabditis elegans* in precise spatiotemporal chemical environments." Nature methods 8.7 (2011): 599-605.) and many others in the field use “odor gradient” and “odor ramp” when, by the reviewer’s view, they should read “odorant gradient” and “odorant ramp.” To address this, we have inspected every instance of “odor” and replaced with odorant when appropriate. Second, we now say that the ORN responds to “both the first derivative of the odorant concentration and the time course of the odorant concentration itself.”

10) Introduction: The authors correctly and cautiously state: "the LNs structure a circuit that putatively implements a bistable inhibitory system". Then they continue that "one state computes odor saliency". What exactly does saliency mean? Do the authors refer to the ability of the stimulus to provoke a behavioral response? If so, this statement should also be toned down as it is a hypothetical assumption. The same applies to the second state. Please separate more clearly your anatomical findings from potential functions of the circuitry.

Saliency with respect to odors is that the odor, as represented by neural circuit activity, dominates said activity at the expense of other odors. The reason that panglomerular inhibition could lead to odor saliency, is that it would dampen weak ORN responses in favor of stronger ones indiscriminately (since the inhibition is panglomerular). If inhibition is not panglomerular, one would not necessarily promote saliency of the strongest odor with respect to the weaker ones. Saliency is produced by contrast-generating mechanisms that amplify small differences into larger ones, and lateral inhibition is known to play this role in the antennal lobe and many other neuropils. We aren’t referring to behavioral responses. Still, we have made it clearer that our conclusions are hypotheses based on the synaptic connectivity and circuit structure.

*11) Results: Please do not refer to the mushroom body as a center for learning and memory in contrast to the lateral horn (see comment 3).*

See our response to comment 3 above.

*12) Results: The authors state that duet LNs play a "far stronger role in postsynaptic inhibition". Actually, the result is that those duet neurons have more synapses onto PNs than the trio LNs. All conclusions beyond that are hypothetical, i.e., how strong the inhibition of the postsynaptic neuron actually is.*

We now make it clearer that our conclusion of a stronger role in postsynaptic inhibition for the Duet is due to the fact that the Duet makes far more synapses onto the dendrites of the uPNs than the Trio does. We state that this may indicate a far stronger role in postsynaptic inhibition. Our initial stronger statement was motivated by the fact that experimental evidence from our labs show that in the larva an excitatory connection with more synapses indicates a stronger connection. We understand that this knowledge is neither widespread nor included in this manuscript, and therefore we adjusted our statement accordingly. In vertebrates it is known that larger synapses, with large synaptic surfaces, are associated with stronger connections. In the larva most synapses are similarly sized as can be seen from electron microscopy, and perhaps someone will eventually demonstrate that most individual morphological synapses have a similar strength.

13) Results: The subheading "Extracting features from odor space at the first synapse" refers to too speculative interpretation of the anatomical data.

We have changed this subheading to be “Multiglomerular System” instead. This better mirrors the subheading “Uniglomerular System” that was used earlier in the manuscript. We feel, though, that this diminishes the summarizing opportunity of subheadings, which is that our interpretation of mPNs is that they extract features from odor space at the first synapse. The combination of multiple ORNs into a single postsynaptic arbor, with the added complexity of LNs mediating lateral interactions among glomeruli, suggests exactly that. If the reviewer does not strongly object, we’d rather keep the prior subheading, which better summarizes our interpretation of the data.

*14) Results: The terms "Choosy LNs" and "Picky LNs" I found somehow confusing. Choosy LNs are GABergic, and Picky LNs mostly glutamatergic, correct? Both names indicate that they selectively innervate only few glomeruli. Maybe one can give them names that are more indicative of their differential connectivity.*

See our response to the essential revisions/review 1 with respect to Picky and Choosy LNs. We have added substantial text clarifying the difference between the two types of LNs.

15) Results: The authors state that "Picky LNs may select for ORNs that are similar in a dimension other than odor preference". I assume the authors mean the array of odorants binding to the receptors ("odorant selectivity" or "odorant binding profile"), because with "odor preference" I intuitively think of the animals' behavior.

We have made this sentence clearer by saying “odorant binding profile,” like the reviewer suggests.

*16) Results: The authors often use the term "feature", and it is not really clear what exactly that means. Perhaps one could replace that term with "combinations of ORN inputs".*

When we say “features,” we are referring to what the combination of ORN inputs putatively mean. What do the inputs have in common? For example, one of the Picky LNs preferentially receives inputs from ORNs that respond to aromatic compounds. A possible feature being extracted would be “aromatic odor” that might be predictive of the presence of food.

17) Results: The subheading "Non-ORN sensory neurons and interactions among LNs can alter the operational state of the olfactory system" is too speculative.

Operational states the authors have not investigated here. And in the entire Results section, I suggest that the authors make clear what they have analyzed (anatomical connectivity) and what is a potential interpretation (functional consequences from the circuitry, e.g., which neuron might provide stronger or weaker inhibition).

For this subheading, we changed the word “can” to “could” to make it clear that what we are presenting is putative. We have also added additional language in this section as well as throughout the entire manuscript to make it clear that the functional roles to the circuits are putative and derived from the connectivity and neurotransmitter signatures that we have presented, and the existing deep literature on olfactory LNs, particularly from the Rachel Wilson lab in adult fly.

*18) Discussion: This section is perfectly clear.*

Thank you.

*19) The reader might wonder why in this anatomical study EM pictures are not included. I suggest that an indication that this work is based on a published EM stack, and that the volume used here is available online, could appear already early in the manuscript. The authors indicate that only in the Materials and methods section.*

We have added an additional figure supplement to Figure 1 that presents EM slices and points out cell types and synapses. We have also created an entire section early in the manuscript that discusses the EM dataset, bringing in information that before was present in the Materials and methods.

Reviewer #3:

In Figure 1—figures supplement 2 ORN synapses are rendered in the same color as the skeleton. This is not very helpful, since it is not clear what is a synapse and what is part of the arbor.

"Therefore Broad LNs may keep the entire olfactory system within the dynamic range to remain responsive to changes in odor intensities." Consider rephrasing. It is not that these neurons keep the system within the dynamic range, but rather these properties define the dynamic range. Same later in the text, when the authors state: "to maintain uPN output within the dynamic range".

"Importantly, Broad LNs also synapse onto each other (Figure 2C, E) like in the adult (Okada et al., 2009), suggestive of a mechanism for sequential recruitment as overall stimulus intensity increases". It is not clear how this would work given that LNs are inhibitory. Please explain.

Figures 2 and 3: order of panel E and D is switched.

It is hard to understand from the text how Choosy LNs that are driven by only a small subset of glomeruli can drive postsynaptic inhibition of most other glomeruli. Inspecting Figure3—figure supplement 1 reveals that Choosy LNs receive ORN input in 9 glomeruli, but synapse on uPNs in 17 glomeruli. This should be explained more clearly.

The reason why we render the ORN synapses in the same color as the ORN skeleton is that if we were to make the synapses all the same color (as we do in some other figures; cyan for postsynaptic, red for presynaptic), the entire antennal lobe, as projected to 2D, would appear as an incomprehensible mess of red and cyan lines. Rendering the synapses in the same color as the skeleton seems to best show the glomerular structure of the antennal lobe, as well as the spatial extent of the glomeruli. Since our tracing method represents neurons as single dots in a slice (the skeleton), showing the synapses as the same color of the ORN best allows us to see the spatial extent of the boutons of the ORN and the ORNs relative location in the antennal lobe for identification, which is the main point of that figure. We will in any case offer the skeletons along with the EM online, accessible with a web browser at the Open Connectome Project website.

We removed the hypothesis about Broad LNs being sequentially recruited since we think that this most needs more physiological experiments to verify it.

Thanks for helping us make the statement about dynamic range more precise. We have rephrased it following your suggestion that the inhibitory LNs define the dynamic range within which PNs respond to odors.

The panels D and E for Figures 2 and 3are not mislabeled, it is just that the figure panels best fit within the constraints of the figure this way.

Multiple changes in language were added to make the distinction between Choosy LNs and Picky LNs clear (see the response to reviewer 1). To address the specific point of this reviewer about Choosy LNs being able to receive inputs from only a few glomeruli put synapse onto the uPNs of most glomeruli, we clarified that Choosy LNs present axons with separate dendritic and axonal arbors. This allows the neuron to a small dendritic arbor that is selective, and a large axonal arbor that targets the uPNs of most glomeruli.